

# The efficacy of computed tomography scanning versus surface scanning in 3D finite element analysis

Andre J. Rowe and Emily J. Rayfield

School of Earth Sciences, University of Bristol, Bristol, United Kingdom

## ABSTRACT

Finite element analysis (FEA) is a commonly used application in biomechanical studies of both extant and fossil taxa to assess stress and strain in solid structures such as bone. FEA can be performed on 3D structures that are generated using various methods, including computed tomography (CT) scans and surface scans. While previous palaeobiological studies have used both CT scanned models and surface scanned models, little research has evaluated to what degree FE results may vary when CT scans and surface scans of the same object are compared. Surface scans do not preserve the internal geometries of 3D structures, which are typically preserved in CT scans. Here, we created 3D models from CT scans and surface scans of the same specimens (crania and mandibles of a Nile crocodile, a green sea turtle, and a monitor lizard) and performed FEA under identical loading parameters. It was found that once surface scanned models are solidified, they output stress and strain distributions and model deformations comparable to their CT scanned counterparts, though differing by notable stress and strain magnitudes in some cases, depending on morphology of the specimen and the degree of reconstruction applied. Despite similarities in overall mechanical behaviour, surface scanned models can differ in exterior shape compared to CT scanned models due to inaccuracies that can occur during scanning and reconstruction, resulting in local differences in stress distribution. Solid-fill surface scanned models generally output lower stresses compared to CT scanned models due to their compact interiors, which must be accounted for in studies that use both types of scans.

# INTRODUCTION

Finite element analysis (FEA) is a computational technique that reconstructs stress, strain, and deformation in solid structures. While initially common in engineering, architecture, and orthopaedic sciences, it is now widely used to assess the biomechanics of the human musculoskeletal system, and in recent years it has been a crucial tool in understanding vertebrate biomechanics and evolution (*Ross, 2005*; *Rayfield, 2007*). FEA has been used in studies of 2D (*Rayfield, 2004*; *Rayfield, 2005a*; *Rayfield, 2005b*; *Pierce, Angielczyk & Rayfield, 2008*; *Pierce, Angielczyk & Rayfield, 2009*; *Fletcher, Janis & Rayfield, 2010*; *Ma et al., 2021*) and 3D structures (*Moreno et al., 2008*; *Bell, Snively & Shychoski, 2009*; *Oldfield et al., 2012*;

Corresponding author
Andre J. Rowe,
andre.rowe@bristol.ac.uk

*Cost et al., 2019*; *Rowe & Snively, 2021*) to assess patterns and magnitudes of stresses and strain in both extant and extinct organisms, as well as suture morphology in the crania of reptiles (*Rayfield, 2005a*; *Rayfield, 2005b*; *Jones et al., 2017*) and mammals (*Bright & Gröning, 2011*; *Bright, 2012*). While studies involving FEA commonly focus on stress and strain occurring in the skull during feeding (*Rayfield, 2007*), studies may also examine the biomechanics of other vertebrate appendages (*Arbour & Snively, 2009*; *Lautenschlager, 2014*; *Bishop et al., 2018*).

FEA is popular in studies of fossil taxa as it is a non-destructive and non-invasive method to study the structural mechanics of extinct organisms. These studies are sometimes conducted using geometrically accurate 3D models which are generated through various techniques, including photogrammetry (*Falkingham, 2012*), computed tomography (CT) scanning and surface scanning (*Rayfield, 2007*). Both CT scanning and surface scanning methods have often been used to study fossil vertebrate morphology with 3D geometric morphometrics (*Friess, 2006*; *Harcourt-Smith et al., 2008*; *Kuzminsky et al., 2016*). Surface scanning has also been used to study locomotion via trackway scanning (*Bates et al., 2008*; *Ziegler et al., 2020*), and to scan immovable museum specimens (*Bates et al., 2009*; *Cunningham et al., 2014*)

## Computed tomography (CT) scanning

CT scans have an extensive history in the medical field (*Power et al., 2016*), but in recent decades they have been commonly used in paleontological (*Haubitz et al., 1988*; *Carlson et al., 2003*; *Racicot, 2016*) and zoological studies (*Copes et al., 2016*; *Poinapen et al., 2017*). They allow for a non-invasive visualization of the interior of biological structures and can be used to generate high resolution tomographic data of bone, fossils, and tissues. These data are used to create 3D models which can facilitate biomechanical modelling, geometric morphometric analyses, or phylogenetic analyses.

CT scanning is a powerful tool in biological studies, as the 3D models generated from the scans can capture both internal and external details with precision (*Rowe et al., 2016*). Image quality in CT scans depends on four basic factors: image contrast, spatial resolution, image noise, and artifacts (*Goldman, 2007*), and can also vary by the size of the specimen being scanned and the type of machine used (*Borasi et al., 1984*). While CT scanning offers many advantages, there are disadvantages relative to surface scanning that must be considered, including high costs (*Fred, 2004*), specimen size limitations, and time spent segmenting the data.

## Surface scanning and photogrammetry

Both surface scanning and photogrammetry are increasingly common digitization techniques that have applications comparable to CT scanning (*Remondino, 2011*). Like CT scanning, both methods are used to generate virtual 3D data that can be valuable in biological studies. Surface scanning and photogrammetry may serve as alternative methods to CT scanning that avoid the expenses and large specimen size restrictions of CT scanning (*Mallison, Hohloch & Pfretzschner, 2009*), though the resulting 3D models lack the internal anatomy of complex structures such as the endocast of the skull (*Sutton, Rahman &*

*Garwood, 2016*). Since surface scans tend to miss intricate details of smaller specimens, *e.g.*, bone textures and teeth, CT scanning is generally the preferred method when dealing with small specimens in palaeobiological studies.

Laser scans and white light scans are two types of surface scans used in biological studies. Laser scanners use a one-dimensional type of scan with a line pattern, which may lead to a high error rate for certain objects (*Persson et al., 2009*). White light scanners use a two-dimensional stripe pattern for obtaining three-dimensional data. Generally, white light scanning is more accurate and faster in the scanning of plaster models in medical studies (*Jeon et al., 2014*); *Peterson & Krippner (2017)* found little difference in the effectiveness of one type of surface scan when comparing the fidelity of 3D printed teeth and osteoderms.

Studies have already investigated which 3D scanning type is more reproducible in medical studies; *Fahrni et al. (2017)* concluded that multi-detector computed tomography (MDCT) led to greater variability in results when compared to three-dimensional surface scanning (3DSS) but noted that more experimentation was necessary to explain their first impression and expand on the results. *Kulczyk et al. (2019)* examined how cone-beam computed tomography (CBCT) scans compare to optical scans when comparing tooth models in 3D printing and found that high-resolution CBCT is a sufficient method to obtain data, but the texture quality was poorer than in optical scan. *Soodmand et al. (2018)* examined the mean model deviation in CT data compared to reference optical 3D scans and found no significant discrepancies in 3D models of a human femur. Other studies have compared 3D models created via photogrammetry and CT scanning in contexts broader than medical studies. *Lautenschlager (2016)* noted that while photogrammetry is the most cost-efficient and easily-reproducible method, it can be limited in its applications due to its inability to capture internal geometries and complex surfaces. *Fahlke & Autenrieth (2016)* similarly noted that CT scanning has its main strength in capturing internal features, but surface scanning was otherwise sufficient in 3D model generation. *Hamm et al. (2018)* concluded that CT scanning was likely the better option for large, complex structures like a *Tyrannosaurus rex* skull, as the data-capture effort of photogrammetry is directly linked to the size and colour of the specimen and to the complexity of its shape; however, this conclusion did not consider costs and size restrictions of some CT scanners. CT scanning is independent of the specimen's shape and complexity, with an accuracy and reproducibility of less than 1% mean error (*Marcus et al., 2008*) which can be advantageous in both time spent acquiring data and the quality of the models.

CT scanning and surface scanning have previously been compared in terms of topography and morphology (*Waltenberger, Rebay-Salisbury & Mitteroecker, 2021*) and the efficiency of several different surface scanning methods have been compared in terms of digitization quality (*Díez Díaz et al., 2021*). However, no study has evaluated the downstream differences in finite element models created from CT scans versus surface scans, or has evaluated the possible discrepancies in 3D finite element results when comparing surface scanned models and CT scanned models derived from the same material. Additionally, no studies have evaluated how to reduce possible discrepancies between results in 3D models generated from different scanning methods. Though the resolution of surface geometry and its influence on FE results has been studied (*McCurry,*

*Evans & McHenry, 2015*), this study is the first to evaluate the use of both CT scans and surface scans in 3D FEA.

## Primary hypotheses and rationale

In this study, we investigated the comparable difference in stress and strain output data between finite element models of the same specimen and loading conditions, created either from white light surface scanning or computed tomography methods. We assessed the FE results from 3D models of three reptile skull specimens: a Nile crocodile (*Crocodylus niloticus*) (Fig. 1), a monitor lizard (*Varanus salvator*) (Fig. 2), and a green sea turtle (*Chelonia mydas*) (Fig. 3). The taxa were chosen for their morphological diversity, differences in feeding biomechanics, and ready availability of muscle data in the literature, including insertions and muscle force components. Crocodilians are noted for their akinetic skull properties due to possessing a secondary palette (*Ferguson, 1981*; *Bailleul & Holliday, 2017*), which provides a contrast to monitor lizards which possess a more flexible, kinetic skull lacking a secondary palette (*Arnold, 1998*; *Herrel et al., 2007*; *Handschuh et al., 2019*). The green sea turtle was chosen as a means of testing a beaked-omnivorous animal (*Arthur, Boyle & Limpus, 2008*; *Nishizawa et al., 2010*) in contrast to sharply toothed carnivores.

Each specimen was digitized using a Nikon XT H 225ST μCT scanner and an Artec3D Space Spider surface scanner. CT parameters were set to 225 kV, 449 mA, 101 W, 1.5 mm copper filter, 0.5 s exposure time, reflection rotating target, 3141 projections, and 4 frames per projection. Manufacturers specifications list the surface scanning 3D point accuracy to 0.05 mm and the 3D resolution at 0.1 mm, but this depends on distance from the specimen to the scanner and specimen size. It is unlikely such resolution was achieved in this study, due to the large specimens needing to be scanned at a certain distance away. The surface scanner was connected to a Dell Alienware 13 Core i7-6500U laptop with 16 GB of RAM for processing complex images. 3D models were created as STL files, because they are simple to work with and supported by the majority of 3D visualization and editing software packages (*Sutton et al., 2001*).

### Null hypotheses (1)

3D stress and strain magnitudes and patterns of stress for both the CT scanned models and surface scanned models will be identical when they are analysed with identical boundary conditions and material properties.

### Alternative hypotheses (2)

3D stress and strain magnitudes and patterns of stress will vary between CT scanned models and surface scanned models when they are analysed with identical boundary conditions and material properties. We predict that surface scanned models experience lower stress and strain due to possessing dense internal geometries that are reconstructed in model editing software, while CT scanned models possess geometrically accurate interiors containing more hollow space.

These hypotheses relate to the stress and strain of 3D skeletal structures when scanned using two different methods. Stress is a physical quantity that expresses the internal forces that neighbouring particles of a material exert on each other, and strain is the measure of the

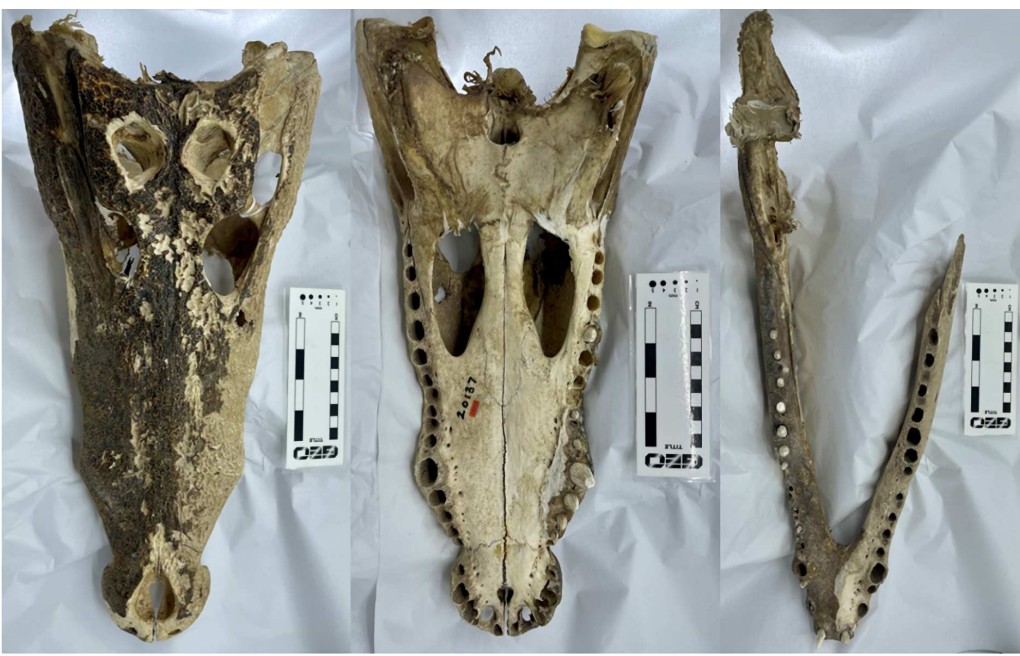

**Figure 1** **The *Crocodylus niloticus* skull (BRSUG 28959) used in the study.** Left to right: cranium in dorsal view, cranium in ventral view, and mandible in dorsal view. Note both the presence of fibrous tissues in the specimen and the broken left ramus in the mandible. Photos by A. Rowe.

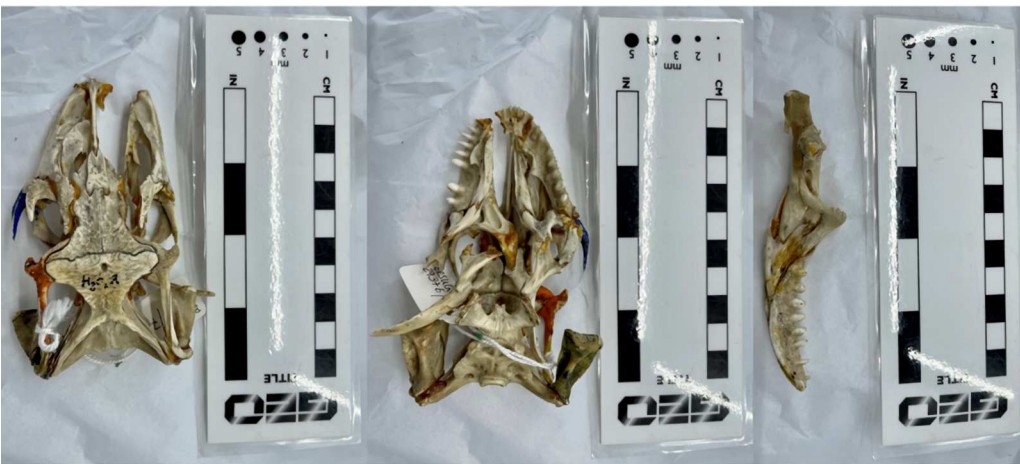

**Figure 2** **The *Varanus salvator* skull (BRSUG 29376/7) used in the study.** Left to right: cranium in dorsal view, cranium in ventral view, and the single left mandibular ramus in medial view. Note the partially broken right maxillary and jugal bones in the skulls. Photos by A. Rowe.

material's deformation when a stress is applied. The skull models were primarily compared by mean von Mises stress (*von Mises, 1913*), a value which accurately predicts how close ductile (slightly deformable/non-brittle) materials like bone are to their failure point. Skull models with lower von Mises stress were judged to be stronger under the imposed bite

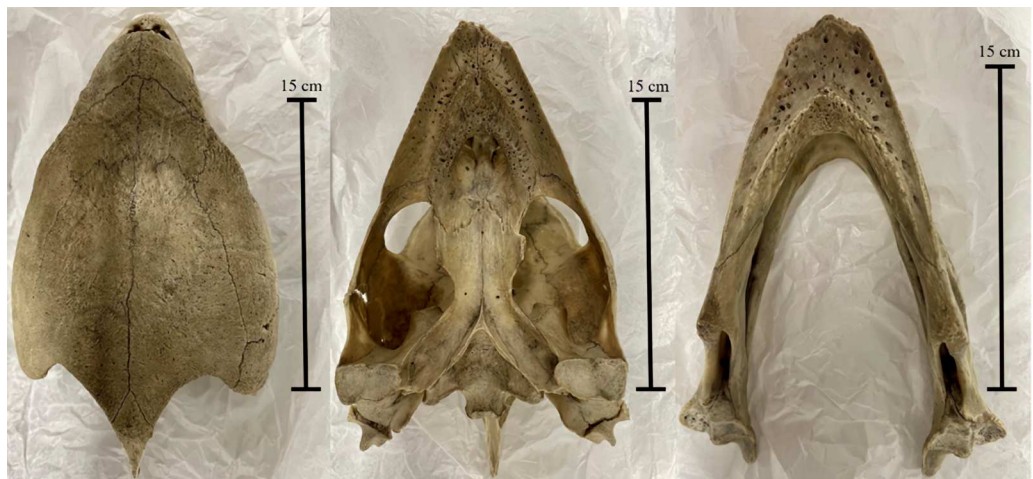

**Figure 3** **The *Chelonia mydas* skull (Ost 160 – Bristol Biological Sciences collection) used in the study.** Left to right: cranium in dorsal view, cranium in ventral view, and mandible in dorsal view. Photos by A. Rowe.

simulations, as lower stresses indicate less susceptibility to breakage or deformation under the imposed load.

## MATERIALS & METHODS

### Scanning procedures

We created 3D models of the crania and mandibles of three phylogenetically disparate taxa using both CT scanning and surface scanning. These specimens include a Nile crocodile (BRSUG 28959), a monitor lizard (BRSUG 29376/7), and a green sea turtle (Ost 160). The reptiles are housed in the University of Bristol Geology collection (BRSUG) or the University of Bristol, School of Biological Sciences teaching collection (Ost/H1b). The Nile crocodile skull was selected for its relatively large size which enabled easier surface scanning, as intricate details including wrinkled textures and teeth are often difficult to capture when scanning small specimens. This specimen possesses fibrous tissues in its cranium and mandible which were captured during CT scanning and surface scanning, which may present a potential issue when surface scanning extant osteological material. This is due to fibrous tissues potentially leading to the creation of unrealistically large surfaces on the model, whereas the CT scanning preserves the intricate details without unrealistically enlarging the surface. The Nile crocodile was missing the posterior part of the left mandible. The *Varanus salvator* specimen was missing the entire right mandible, and the right maxilla and jugal were displaced from the cranium.

All reptile specimens were scanned using a Nikon XT H 225ST µCT housed in the Life Sciences Building, Bristol, UK. Due to the size of the adult crocodile skull, the cranium and mandible were scanned separately, while the turtle and monitor lizard were scanned with both the crania and mandibles held together by foam. CT scanning crania and mandibles together, as was done with the turtle and lizard, was not an issue due to all internal details

being captured separately. Additionally, the segmentation process post CT scanning can separate intertwining models. All specimens were scanned at 120 μm. The TIFF files were imported into Avizo Lite version 9.7 at voxel dimensions 1-1-1 to match the native scan resolution and then segmented using only the Threshold tool. CT scanned models were scaled in MeshLab 2020.03 to adjust length and width dimensions to their surface scanned counterparts as needed. These models were then exported as the STL file type.

The same CT scanned individuals were surface scanned using an Artec Space Spider handheld scanner. The scans were made at 7–8 frames per second, with the 'real-time fusion' option enabled. Real-time fusion aids in piecing together scans during the scanning process and may save time when building the full model in Artec Studio Professional 14. Crania and mandibles were all scanned separately and created as separate 3D object files to avoid both large file sizes and intertwining crania and mandibles during surface scanning.

Surface scans were imported into Artec Studio 14 Professional where sections of scanned skulls were oriented together, registered, and then merged into a single object. Stray pixels were deleted, as well as frames with maximum error values above 0.3. Once we were satisfied with the alignment of the individual scans, we applied Global Registration to convert all one-frame surfaces to a single coordinate system using information on the mutual position of each surface pair. We were satisfied with our alignment once the scans were free of floating pixels and generally resembled the bones we scanned. We then applied a sharp fusion to create a polygonal 3D model, which solidifies the captured and processed frames into an STL file. We used Sharp Fusion rather than Fast Fusion or Smooth Fusion as it best preserves fine details of the scans, including small teeth and rugose bone textures, which were present in the crocodile skull. We then used the small-object filter to clean the STL file of any remaining floating pixels, which are inevitable in most surface scanning procedures. Additionally, we used the fix holes function to fill any open areas (Fig. 4). The STL files were then exported from Artec Studio 14 Professional and imported into Blender version 2.82 for both surface editing and reconstruction of missing elements in the case of the crocodile and lizard mandibles, as well as the missing cranial elements of the lizard.

### 3D finite element model editing

Blender 2.82 was used for more precise editing, typically using the Sculpt functions to smooth over any unnatural-looking surfaces or creases that tend to appear in surface-scanned models. Most CT-scanned models did not require extensive editing; they were run under Geomagic's Mesh Doctor function to remove self-intersections which often resulted after segmentation in Avizo. Due to surface scanning producing hollow models, it was necessary to import the hollow models into Avizo Lite 9.7 and segment them to achieve results comparable to the CT models. This was done by converting the STL files into TIF files, segmenting the interior of the model using Avizo's threshold tool, and generating a surface which was then exported as an STL. This process fills in the entirety of the surface scanned models, leading to a denser interior than that of the trabecular bone preserved in CT scans. We worked under the assumption that the dense interior functions more similarly to the CT scanned models than leaving the models hollow, as stated in our alternative hypothesis (2).

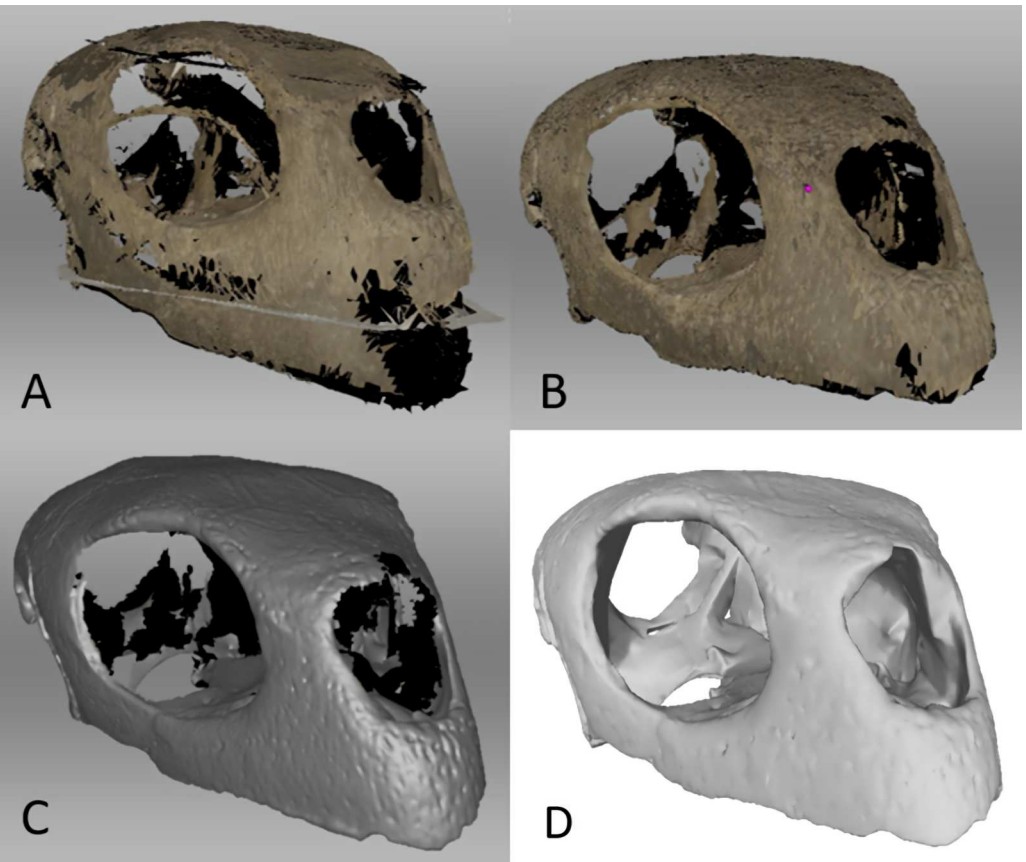

**Figure 4** **Surface scanned *Chelonia* cranium.** (A) Prior to Global Registration in Artec Studio 14 Professional, (B) after Global Registration and outlier removal in Artec Studio 14 Professional, (C) after Sharp Fusion in Artec Studio 14 Professional, which converts the scans into an STL file, and (D) the same STL file in MeshLab 2020.06 after surface editing in Artec Studio 14 Professional and Blender 2.82 to close gaps in the model and better match the geometry of the CT files.

The interior details of surface-scanned crania are generally not captured during scanning. In this study, inner cranial details were constructed from observations of the CT scanned models. We did this using the Sculpt function in Blender 2.82 for the turtle cranium, due to its dense skull (Fig. 4). This included the parietal and postorbital bones of the turtle, as they were difficult to capture during surface scanning.

For the crocodile mandible, the posterior end of the left mandibular ramus was missing (Fig. 1). This was fixed in Blender 2.82 by deleting the missing left mandibular ramus at the middle point of the mandible and duplicating the right mandibular ramus. The right ramus was then mirrored and reattached at the mandible's anterior to create a complete mandible. The right ramus of the monitor lizard mandible was also missing (Fig. 2), and an identical procedure using the left rami was applied to generate a complete mandible. Additionally, the left maxillary and jugal bones of the lizard's cranium were missing, and an identical duplication and mirroring approach was used. This procedure was used for both the surface scanned models and the CT scanned models to best achieve identical geometries

Peerj

Table 1 Number of elements and traingles in each model tested.

| Specimen name | Cranial elements (CT) | Cranial elements (surface scan) | Mandible elements (CT) | Mandible elements (surface scan) | Cranial triangles (CT) | Cranial triangles (surface scan) | Mandible triangles (CT) | Mandible triangles (surface scan) |
|---|---|---|---|---|---|---|---|---|
| Nile crocodile (*Crocodylus niloticus*) | 1,386,928 | 1,554,868 | 2,019,264 | 1,801,958 | 167,940 | 217,306 | 1,386,928 | 247,090 |
| Monitor lizard (*Varanus salvator*) | 159,358 | 172,470 | 688,656 | 135,450 | 159,358 | 172,470 | 588,656 | 135,450 |
| Green sea turtle (*Chelonia mydas*) | 90,188 | 246,798 | 236,498 | 197,722 | 90,188 | 246,798 | 263,498 | 197,722 |

for FE testing and avoid inconsistencies as much as possible. The merging of duplicated geometries in the models resulted in intersecting triangles, which generally causes meshing procedures to fail when creating finite element meshes. This was fixed by importing models containing self-intersections into MeshLab, deleting intersecting triangles, and then using the hole-filling function in Geomagic Studio 12 to replace missing triangles.

Once our models were free of holes and intersecting triangles, the 3D models were imported into Geomagic Studio 12. The Mesh Doctor tool was then selected, which corrects intersecting triangles, sharp edges, and holes, thus reducing the likelihood of errors when meshing the models. The remesh tool was used to help correct irregularly sized triangles in each model. Both element and triangle counts were reduced using the decimate tool as to both shorten analysis times in Abaqus (Table 1) and to aid in reducing intersecting triangles and sharp edges, which are more common in high-element STLs. Volume and surface area for each 3D model was recorded (Table 2).

The models were exported from Geomagic and imported into HyperMesh (Altair) as four-noded tetrahedral elements. Properties were assigned to the various materials, including Young's modulus, the material's stiffness, and Poisson's ratio, the deformation of the material in directions perpendicular to the direction of loading. Alligator skull bone properties (*Zapata et al., 2010*; *Porro et al., 2011*) were assigned to both the crocodile and turtle (Table 3). Alligator bone has been used previously as an extant analogue in turtle studies (*Ferreira et al., 2020*) and was thus considered acceptable here. Lizard bone properties (*Dutel et al., 2021*) were assigned to the monitor lizard (Table 3). All materials were treated as isotropic and homogeneous. As the main purpose of the study was to compare differences in stress and strain results due to geometry, it was considered acceptable to use these material property values.

Constraints were assigned at anterior tooth edges and at the beak in the case of the turtle skull to simulate feeding loads. We chose anterior feeding constraints for each model rather than posterior to best visualize von Mises stress occurring all throughout each model for comparative purposes. Constraints were assigned to the hinges of the articular and quadrate to prevent the model from freely floating. Three constraint points were selected per quadrate and articular hinge for each model. Three degrees of freedom were selected for each analysis at X, Y, and Z. The number of constraint points, typically three per tooth or beak, were kept consistent for each taxon and type of scan (Fig. 5).

Once satisfied with the constraint selection, these models were imported into Abaqus to determine stress and strain in the crania and mandibles of the models. Muscle locations

**Table 2** Volume and surface area of each model tested: volume is in cubic millimeters, and surface area is in square millimeters.

| Specimen name | Cranial volume (mm³) (CT) | Cranial volume (mm³) (surface scan) | Mandible volume (mm³) (CT) | Mandible volume (mm³) (surface scan) | Cranial surface area (mm²) (CT) | Cranial surface area (mm²) (surface scan) | Mandible surface area (mm²) (CT) | Mandible surface area (mm²) (surface scan) |
| --- | --- | --- | --- | --- | --- | --- | --- | --- |
| Nile crocodile (*Crocodylus niloticus*) | 1256734.25 | 1716394.12 | 944250.68 | 1150979.75 | 346104.23 | 229887.71 | 285409.23 | 170483.37 |
| Monitor lizard (*Varanus salvator*) | 23561.33 | 24318.68 | 7452.24 | 10201.48 | 22301.37 | 16554.88 | 8863.39 | 6507.72 |
| Green sea turtle (*Chelonia mydas*) | 153623.06 | 173715.40 | 37029.62 | 34135.75 | 77998.11 | 67361.12 | 16474.20 | 14943.83 |

**Table 3** Material properties applied to 3D models.

| Specimen name | Young's modulus (GPa) | Poisson's ratio |
| --- | --- | --- |
| Nile crocodile (*Crocodylus niloticus*) | 15 | 0.29 |
| Green sea turtle (*Chelonia mydas*) | 20.49 | 0.4 |
| Monitor lizard (*Varanus salvator*) | 22.8 | 0.3 |

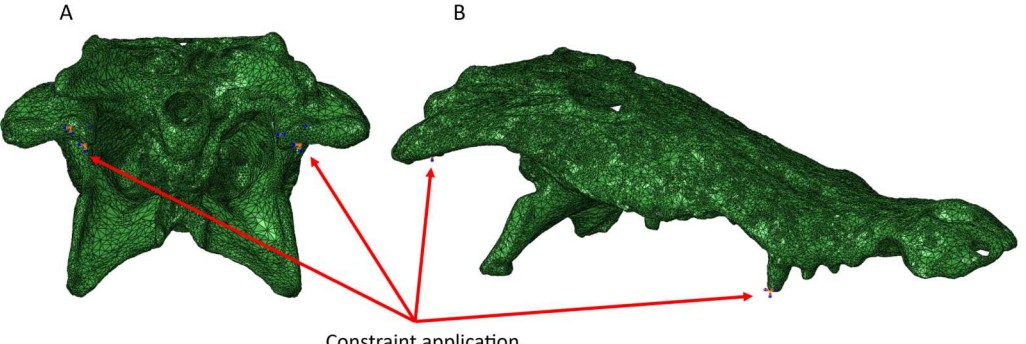

**Figure 5** **Areas of constraint application in the *Crocodylus* CT scanned cranium (A) in posterior view and (B) lateral view.** Three constraints were applied to each side of the quadrate to prevent the model from floating in space, and an additional constraint was applied to the anterior teeth to simulate contact with a food object. For mandible models, three constraints were applied to the posterior hinge of each articular bone. Identical constraint protocol was followed for each reptile model.

and the nodes selected to represent muscle attachment and insertion were based on reconstructions of muscle anatomy from *Holliday (2009)* for *Crocodylus* (Fig. 6) and *Varanus* (Fig. 7) and *Jones et al. (2012)* for *Chelonia* (Fig. 8) (Table 4). Each muscle body was assigned a local coordinate system to simulate the direction of pull of the muscles on the crania and mandibles. A single coordinate system per muscle was created. Muscle force components applied to the model were calculated by dividing muscle force (N) by number of nodes selected per muscle (see Supplementary Information).

Muscle force values were obtained from previous studies involving taxa that are phylogenetically related to those used in this study, including *Alligator mississippiensis*

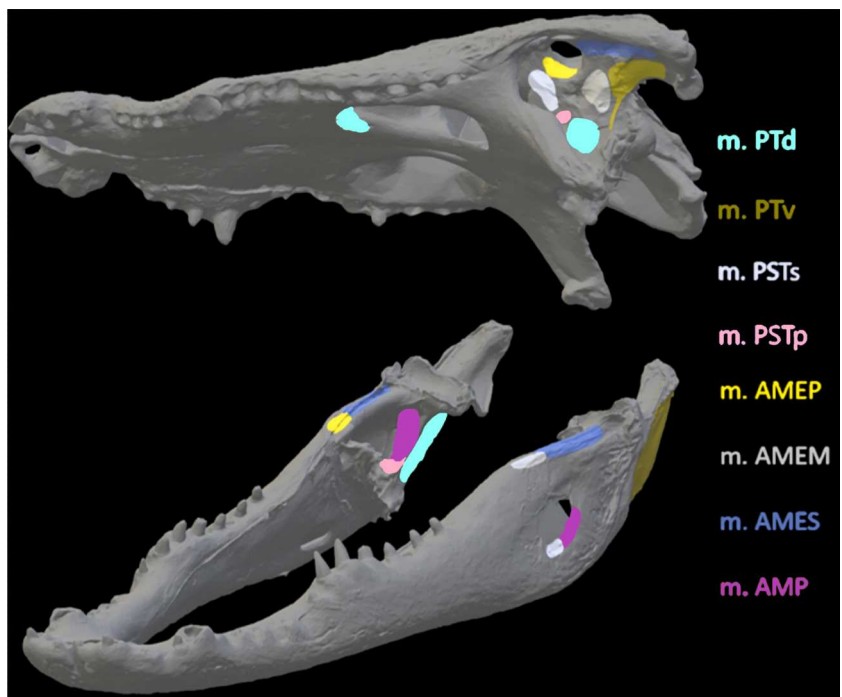

**Figure 6** Muscle insertions where nodes were mapped for the *Crocodylus* model in Abaqus based on *Holliday (2009)*. Nodes were mapped as similarly as possible for both CT scanned and surface scanned models by first applying nodes to CT models and then using the CT models as references when applying nodes to surface scanned models. Muscle abbreviations for all models: mPT, M. pterygoideus; mPSTs, M. pseudotemporalis superficialis; mPSTp, M. pseudotemporalis profundus; mAMEP, M. adductor mandibulae externus profundus; mAMEM, M. adductor mandibulae externus medialis; mAMES, M. adductor mandibulae externus superficialis; mPTd, M. pterygoideus dorsalis; mAMP, M. adductor mandibulae posterior; mPRp, M. adductor mandibulae internus Pars pterygoideus; mAP, M. adductor mandibulae externus Pars superficialis lateral head.

(*Porro et al., 2011*); see Supplementary Information) applied to *Crocodylus niloticus* and *Varanus niloticus* (*Dutel et al., 2021*) applied to *Varanus salvator*. *Platysternon* muscle force values were chosen as a proxy for *Chelonia mydas* due to possessing the highest recorded values of extant turtles which may align more closely with the relatively large *Chelonia* skull (*Ferreira et al., 2020*); S. Lautenschlager, pers. comm. 2021). Once all constraints and nodes were applied across CT scanned and surface scanned models, FE analyses were run under linear static assumptions, with unchanging loads and material properties in the software Abaqus (Simulia). Stresses were compared using von Mises stress, which is used to predict failure under ductile fracture, or fracture characterized initially by plastic deformation, commonly occurring in the bone. Stresses were superimposed on the models as contours with a user-specified range of colours to indicate where stresses experienced are least and most substantial, with warmer colours such as red and white signifying high stress, and cooler colours like blue and green representing low stress.

Additionally, we analysed von Mises stresses and deformation occurring at specific points on the models. This was done by plotting ten points at similar locations on each CT

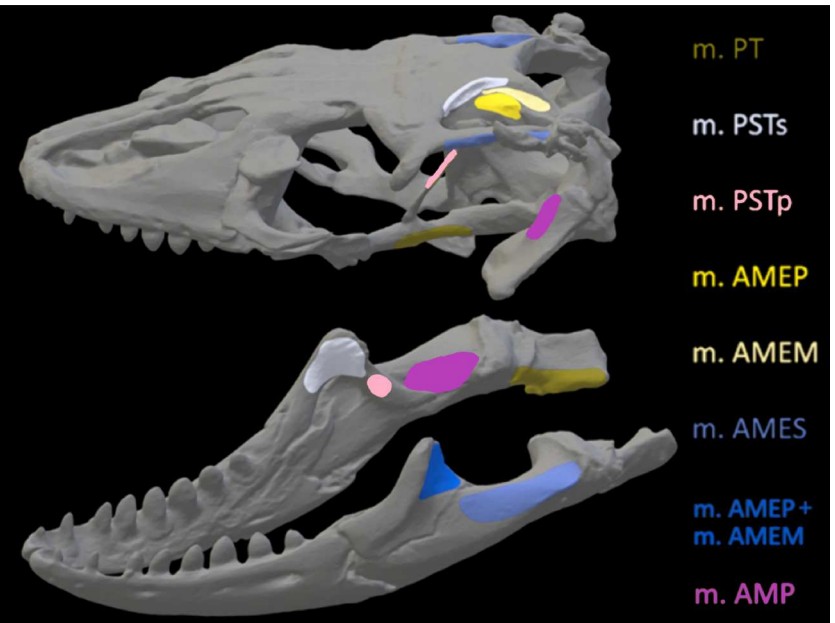

**Figure 7 Muscle insertions where nodes were mapped for the *Varanus* model in Abaqus based on *Holliday (2009)*.**

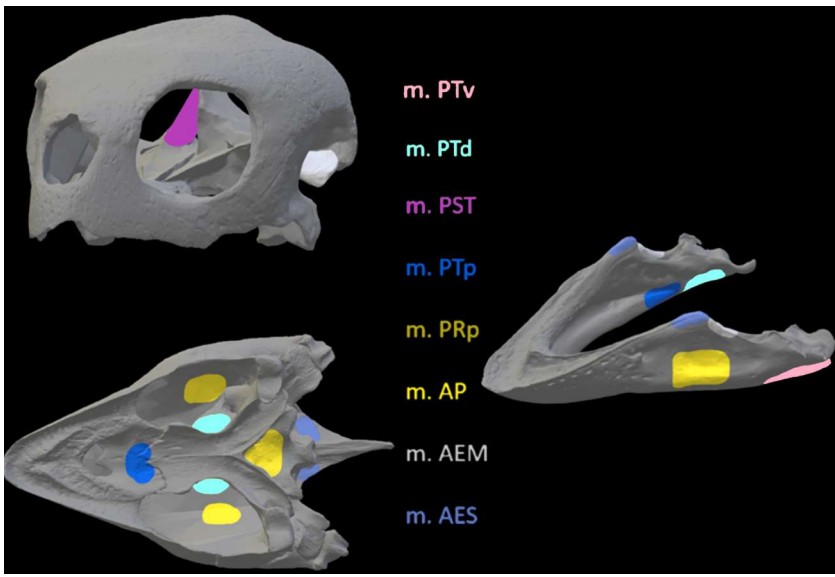

**Figure 8 Muscle insertions where nodes were mapped for the *Chelonia* model in Abaqus based on *Jones et al. (2012)*.**

scanned model and its corresponding surface scan model. We then selected five nodes per point on each model and calculated the mean von Mises stress value at each point (Tables 5, 6 and 7). This was done to better understand stresses occurring at specific points on each

**Table 4  Number of constraints and muscle nodes applied to each model.**

| Crocodylus | Total constraints at quadrate/ articular | m. PTd | m. PTv | m. PSTs | m. PSTp | m. AMEP | m. AMEM | m. AMES | m. AMP |
|---|---|---|---|---|---|---|---|---|---|
| CT cranium | 6 | 16 | – | 27 | 10 | 28 | 14 | 27 | 27 |
| CT mandible | 6 | 54 | 52 | 32 | 32 | 47 | 38 | 62 | 80 |
| Surface scan cranium | 6 | 16 | – | 23 | 10 | 24 | 14 | 28 | 24 |
| Surface scan mandible | 6 | 49 | 51 | 30 | 28 | 27 | 41 | 60 | 70 |

| Varanus | Total constraints at quadrate/ articular | m. PT | m. PSTs | m. PSTp | m. AMEP | m. AMEP + m. AMEM | m. AMES | m. AMP |
|---|---|---|---|---|---|---|---|---|
| CT cranium | 6 | 32 | 16 | 18 | 18 | 16 | 32 | 30 |
| CT mandible | 6 | 42 | 32 | 10 | 18 | 16 | 38 | 34 |
| Surface scan cranium | 6 | 28 | 16 | 20 | 20 | 16 | 32 | 36 |
| Surface scan mandible | 6 | 40 | 31 | 10 | 26 | 26 | 36 | 38 |

| Chelonia | Total constraints at quadrate/ articular | m. PTv | m. PTd | m. PST | m. PTp | m. PRp | m. AP | m. AEM | m. AES |
|---|---|---|---|---|---|---|---|---|---|
| CT cranium | 6 | – | 12 | 12 | 6 | 8 | 12 | 12 | 10 |
| CT mandible | 6 | 20 | 20 | – | 16 | – | 24 | 16 | 20 |
| Surface scan cranium | 6 | – | 14 | 12 | 6 | 10 | 14 | 12 | 14 |
| Surface scan mandible | 6 | 20 | 22 | – | 16 | – | 22 | 16 | 20 |

**Table 5  Mean von Mises stress values (GPa) at locations 1–10 on Crocodylus FE models.**

| Point | Crocodylus cranium CT scan stress | Crocodylus cranium surface scan stress | Crocodylus mandible CT scan stress | Crocodylus mandible surface scan stress | Crocodylus cranium model stress difference | Crocodylus cranium model stress difference percentage | Crocodylus mandible stress difference | Crocodylus mandible stress difference percentage |
|---|---|---|---|---|---|---|---|---|
| 1 | 0.068 | 0.093 | 0.09 | 0.233 | −0.025 | 31.06% | −0.143 | 88.54% |
| 2 | 0.063 | 0.013 | 0.095 | 0.112 | 0.05 | 131.58% | −0.017 | 16.43% |
| 3 | 0.065 | 0.016 | 0.121 | 0.111 | 0.049 | 120.99% | 0.01 | 8.62% |
| 4 | 0.021 | 0.003 | 0.136 | 0.133 | 0.018 | 150% | 0.003 | 2.23% |
| 5 | 0.048 | 0.012 | 0.145 | 0.137 | 0.036 | 120% | 0.008 | 5.67% |
| 6 | 0.041 | 0.028 | 0.158 | 0.095 | 0.013 | 37.68% | 0.063 | 49.80% |
| 7 | 0.036 | 0.039 | 0.033 | 0.056 | −0.003 | 8% | −0.023 | 51.69% |
| 8 | 0.026 | 0.006 | 0.053 | 0.036 | 0.02 | 125% | 0.017 | 38.20% |
| 9 | 0.027 | 0.031 | 0.038 | 0.024 | −0.004 | 13.79% | 0.014 | 45.16% |
| 10 | 0.062 | 0.077 | 0.151 | 0.183 | −0.015 | 21.58% | −0.032 | 19.16% |

**Table 6  Mean unscaled displacement values in cm at locations 1–10 on *Crocodylus* FE models.** Mean values were calculated by recording and averaging five unscaled displacement values at each location.

| Point | *Crocodylus* cranium CT displacement | *Crocodylus* cranium surface scan displacement | *Crocodylus* mandible CT displacement | *Crocodylus* mandible surface scan displacement | *Crocodylus* cranium model displacement difference | *Crocodylus* cranium model displacement difference percentage | *Crocodylus* mandible displacement difference | *Crocodylus* mandible displacement difference percentage |
|---|---|---|---|---|---|---|---|---|
| 1 | 0.00289 | 0.00248 | 0.0412 | 0.0124 | 0.0004 | 15.27% | 0.0288 | 107.46% |
| 2 | 0.00384 | 0.00244 | 0.0553 | 0.0145 | 0.0014 | 44.59% | 0.0408 | 116.91% |
| 3 | 0.00430 | 0.00329 | 0.0861 | 0.0143 | 0.0010 | 26.61% | 0.0718 | 143.03% |
| 4 | 0.00525 | 0.00405 | 0.0936 | 0.0080 | 0.0012 | 25.81% | 0.0856 | 168.61% |
| 5 | 0.00448 | 0.00326 | 0.0509 | 0.0048 | 0.0012 | 31.52% | 0.0461 | 165.53% |
| 6 | 0.00347 | 0.00271 | 0.0214 | 0.0050 | 0.0008 | 24.60% | 0.0164 | 124.86% |
| 7 | 0.00276 | 0.00245 | 0.0341 | 0.0051 | 0.0003 | 11.90% | 0.0290 | 147.87% |
| 8 | 0.00495 | 0.00376 | 0.0316 | 0.0099 | 0.0011 | 27.32% | 0.0217 | 104.51% |
| 9 | 0.00384 | 0.00328 | 0.0263 | 0.0195 | 0.0006 | 15.73% | 0.0068 | 29.69% |
| 10 | 0.00290 | 0.00325 | 0.0192 | 0.0109 | −0.0004 | 11.38% | 0.0083 | 55.15% |

CT scanned model and its corresponding surface scanned counterpart. A similar method was applied to each point where an unscaled mean displacement was calculated by selecting five nodes. This method revealed the amount of deformation occurring in each model and to what quantitative extent each CT scanned model was deforming when compared to the surface scanned models. Points were chosen to capture both as many different bones of each skull as possible and to quantify deformation in both areas of low and high stress on the FE heatmaps.

Once we calculated mean von Mises stress values for all models, we also calculated the mesh-weighted arithmetic mean (MWAM) von Mises stress value for each model using R (*R Core Team, 2021*). This method accounts for element size differences within non-uniform meshes and has been used in previous biomechanical studies of vertebrate palaeobiology (*Marcé-Nogué et al., 2016*; *Morales-García et al., 2019*; *Ballell & Ferrón, 2021*). It can reduce discrepancies in von Mises stress between CT scanned models and surface scanned models. The code is as follows:

## RESULTS

In most FE models, mean von Mises stress magnitudes were generally higher in the CT scanned models than the surface scanned models. The CT scanned models which produced von Mises stresses higher than the surface scanned models were the *Crocodylus* cranium and mandible, *Varanus* cranium, and *Chelonia* cranium. The mean von Mises stresses differed overall by 85.76% between both types of models (Fig. 9), though certain models differed highly while others were comparable in their results, such as the *Chelonia* mandibles. We also calculated the median von Mises stress values for each model (Fig. 10). Median von Mises stress values did vary from the mean stress values, in that the *Varanus*

```
Stressfile<-read.table("model_smises.txt",header = T)
Stressfile
Volumefile<-read.table("model_evol.txt",header=T)
Volumefile
Stress<-Stressfile$SMises
Stress<-as.numeric(Stress)
length(Stress)
Volume<-Volumefile$Evol
Volume<-as.numeric(Volume)
length(Volume)
StressVolume<-numeric(length = length(Stress))
for (i in 1:length(Stress)) {StressVolume[i]<-
Stress[i]*Volume[i]}
MWAM<-SumArea<-
mean(StressVolume)/mean(Volume)
```

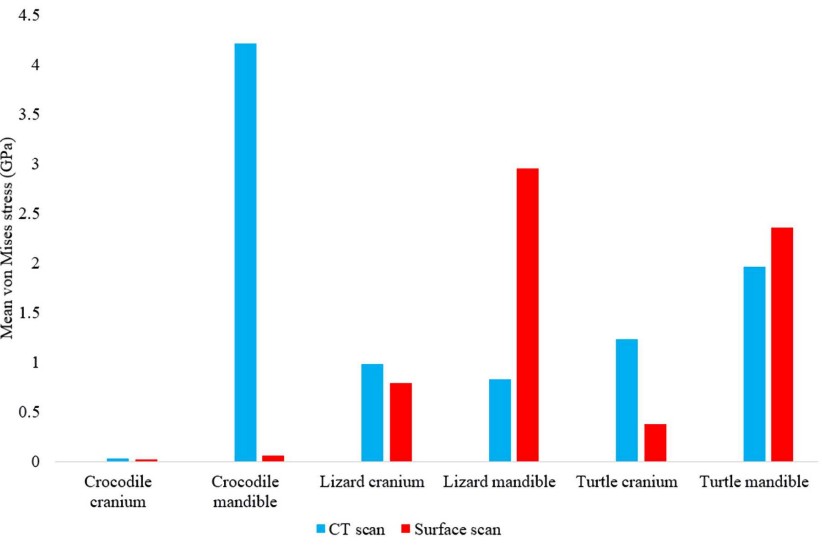

**Figure 9** **Mean unweighted von Mises stress values (GPa) in each FE model.** Note the large discrepancies in the models requiring more reconstructive work, *i.e.*, the crocodile and monitor lizard mandibles.

and *Chelonia* cranial stresses were slightly higher in the surface scanned models. Mean maximum principal strain values similarly differed overall by 86.04% (Fig. 11).

Our data on stress, strain and deformation values at specific points in both models was consistent overall with our mean von Mises stress data for each model in that the data demonstrated comparative trends between models, despite differences in model topography. Similarly, our specific point analysis of the unscaled displacement values yielded consistent results, with models that had undergone extensive reconstructions differing the most in unscaled displacement and those with little reconstruction yielding comparative FE data.

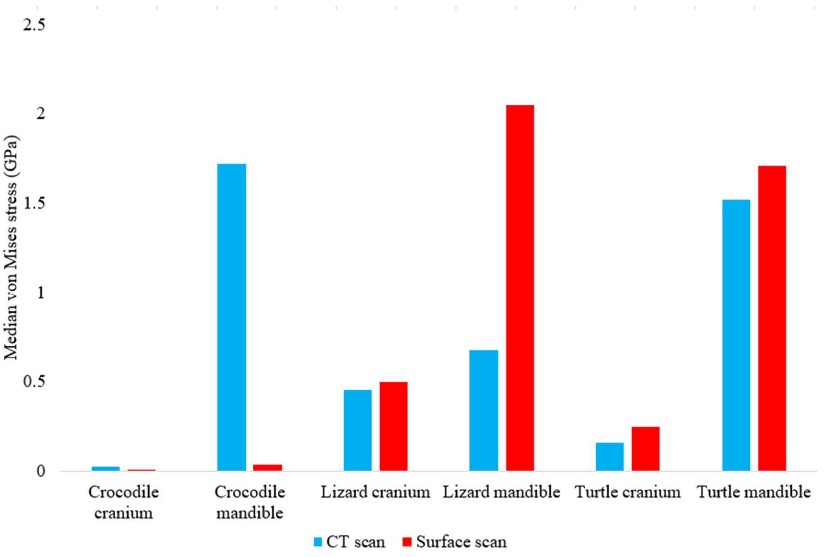

**Figure 10  Median unweighted von Mises stress values (GPa) in each FE model.**

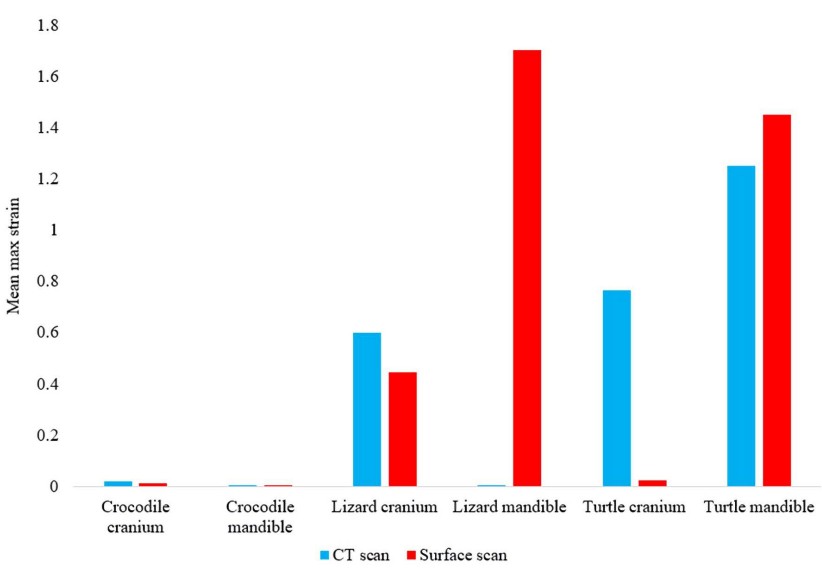

**Figure 11  Maximum principal strain values (Emax) in each FE model.** *Y*-axis represents the strain percentage.

Calculating the mesh-weighted arithmetic mean (MWAM) values greatly reduced the incongruity between CT scanned and surface scanned model stresses, as they differed overall by an average of 35.55% (Fig. 12) compared to the unweighted average 85.76% value.

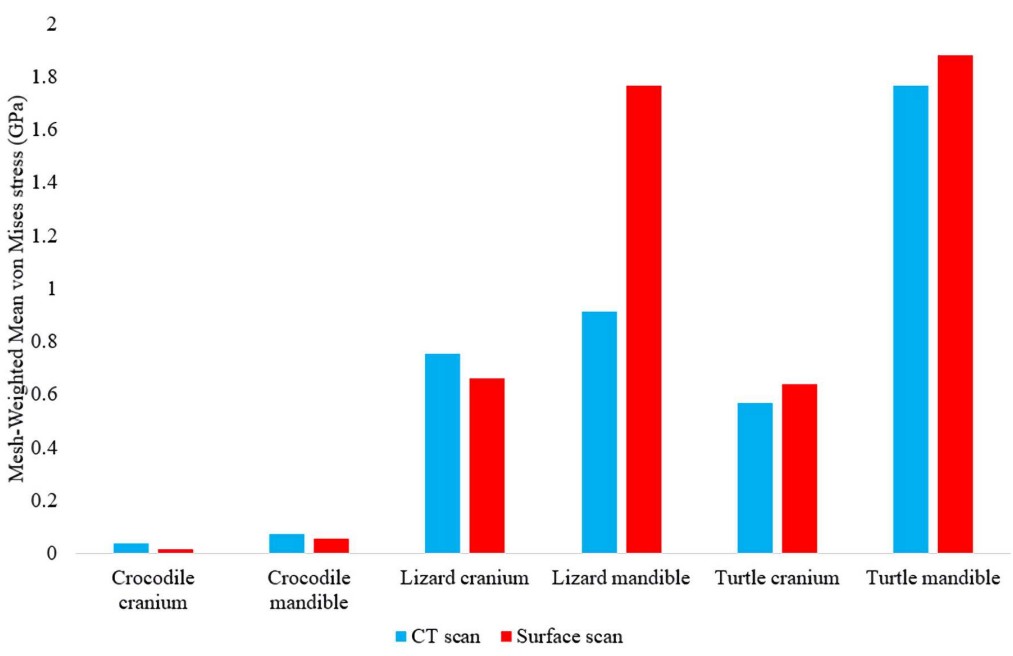

**Figure 12** **Mesh-weighted arithmetic mean (MWAM) von Mises stress values in each FE model.** Note the lower discrepancies in von Mises stress values between model types when the MWAM is calculated.

### *Crocodylus* results

The *Crocodylus* crania were two of the more consistent models in terms of surface geometries, von Mises stress results, deformation, and 3D model properties (Tables 1 and 2). The number of elements between the two model types differed by 11.42%. Mean unweighted von Mises stress differed by 61.37% and mesh-weighted von Mises stress differed by 82.62%. Maximum principal strain differed by 52.12%. Like many of our models, stress distributions were noted for appearing similar in both versions, despite stress magnitudes being inconsistent (Fig. 13). Both models were deforming in similar ways as well (Fig. 14); anterior torsion occurred in each model due to teeth and their constraints only present on the right maxilla. Our specific point mean von Mises stress values overall differed by 75.97% (Table 5) and the mean unscaled displacement values overall differed by 15.27% (Fig. 15; Table 6).

Stress, strain and deformation magnitudes in the *Crocodylus* mandible surface scan model highly deviated from its CT scanned counterpart. Mean unweighted von Mises stress differed by 194.43% and mesh-weighted von Mises stress differed by 23.33%. Max strain differed by 32.6%. Our specific point mean von Mises stress values overall differed by 32.55% (Table 5) and the mean unscaled displacement values overall differed by 114.51% (Fig. 15; Table 6). However, like the *Crocodylus* cranium, stress distributions still appear consistent between the models, despite the stark contrast in mean von Mises stress magnitudes and differences in the topography of the models (Figs. 13 and 15).
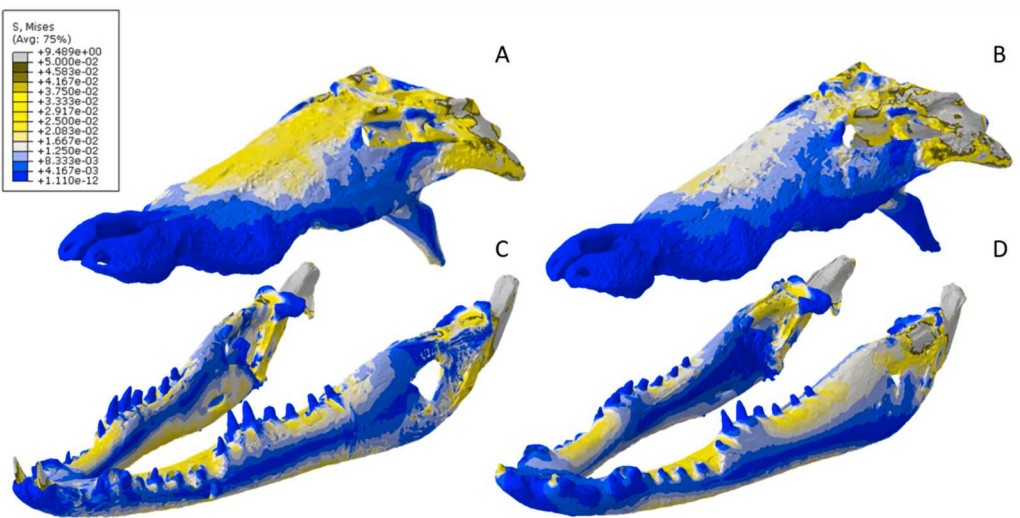

**Figure 13 Von Mises stress results for the *Crocodylus* models.** (A) CT scanned cranium, (B) surface scanned cranium, (C) CT scanned mandible, (D) and surface scanned mandible. Cooler colors like blue indicate low stress occurrences, while hotter colors such as orange indicate higher stresses. All FE model images were scaled to the same maximum stress values for consistency.

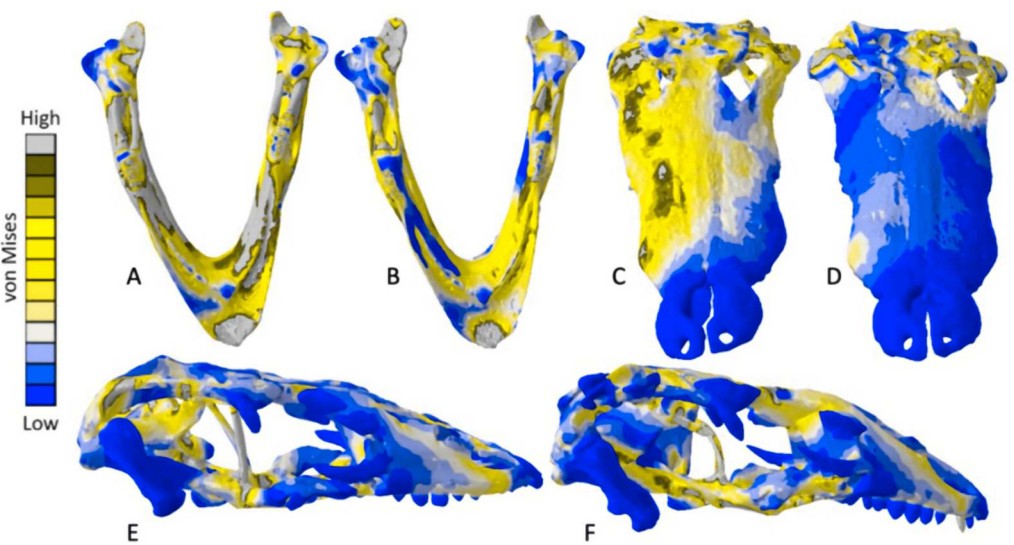

**Figure 14 Exaggerated strain (Emax) deformation results in a selection of FE models tested.** (A) CT scanned *Chelonia* mandible, (B) surface scanned *Chelonia* mandible, (C) CT scanned *Crocodylus* cranium, (D) surface scanned *Crocodylus* cranium, (E) CT scanned *Varanus* cranium, (F) and surface scanned *Varanus* cranium. Magnification was at 75%. Models not to scale. Von mises stress key indicative of high and low values but not to scale across all models.

## *Varanus* results

The *Varanus* crania were two of the most consistent models in their geometries, von Mises stress distributions, and deformation (Fig. 16). Mean unweighted von Mises stress differed

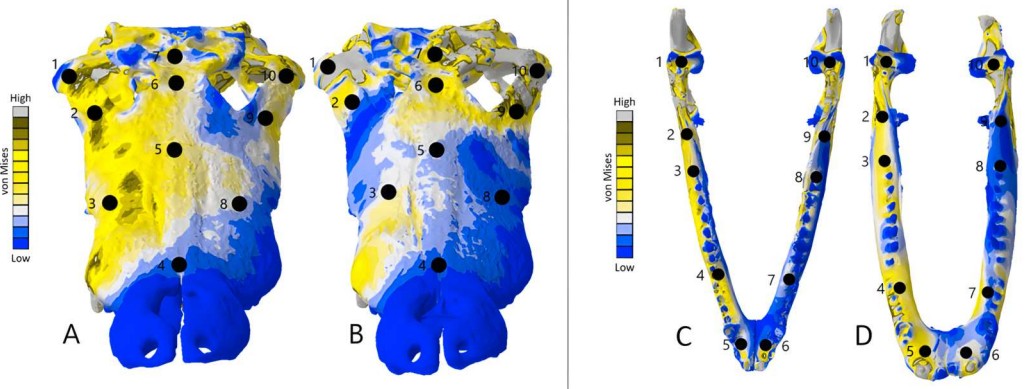

**Figure 15** **Dorsal view of the *Crocodylus* CT scanned cranium (A) and mandible (B) and surface scanned cranium (C) and mandible (D).** The mean von Mises stress of five nodes was recorded at each location, averaged, and documented in Table 8. Both FE model images were scaled to the same maximum stress values for consistency. The mean unscaled displacement of five elements to represent deformation was recorded at each point, averaged and documented in Table 7.

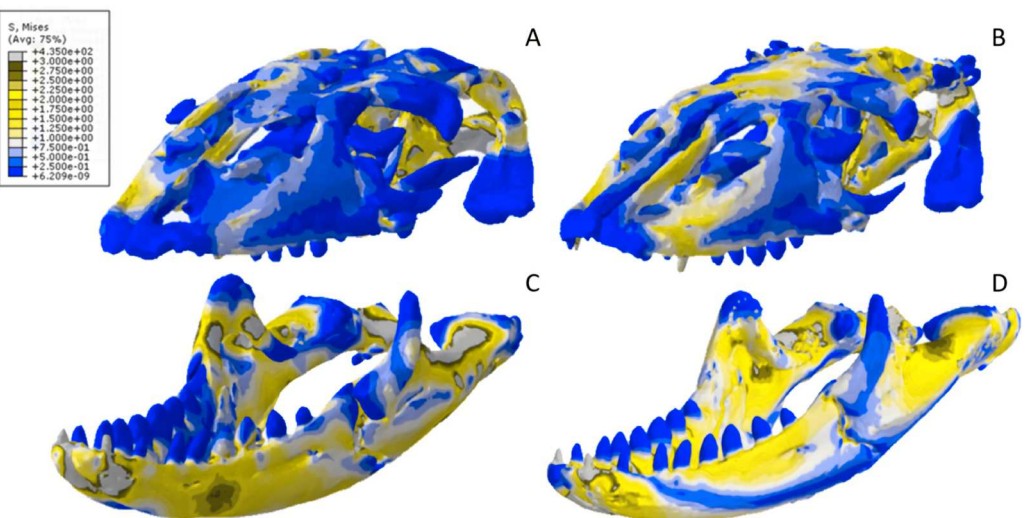

**Figure 16** **Von Mises stress results for the *Varanus* models.** (A) CT scanned cranium, (B) surface scanned cranium, (C) CT scanned mandible, and (D) surface scanned mandible. All FE model images were scaled to the same maximum stress values for consistency.

by 21.14% and mesh-weighted von Mises stress differed by only 3.16%. Mean maximum principal strain differed by 29.5%. Our specific point mean von Mises stress values overall differed by 83.76% (Table 7) and the mean unscaled displacement values overall differed by 24.52% (Fig. 17; Table 8). Deformation was more noticeable in the surface scanned model, especially in the bones of the cranium that were not as dense as the CT scanned model (Fig. 14).

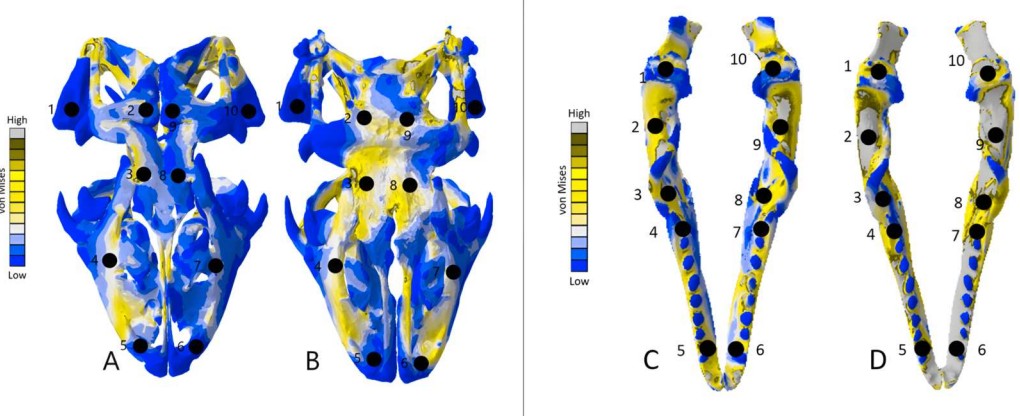

**Figure 17 Dorsal view of the *Varanus* CT scanned cranium (A) and mandible (B) and surface scanned cranium (C) and mandible (D).** The mean von Mises stress of five elements was recorded at each location, averaged, and documented in Table 9. The mean unscaled displacement of five elements to represent deformation was recorded at each point, averaged and documented in Table 10. FE model images were scaled to the same maximum stress values for consistency.

**Table 7 Mean von Mises stress values (GPa) at locations 1-10 on *Varanus* FE models.**

| Point | *Varanus* cranium CT scan stress | *Varanus* cranium surface scan stress | *Varanus* mandible CT scan stress | *Varanus* mandible surface scan stress | *Varanus* cranium model stress difference | *Varanus* cranium model stress difference percentage | *Varanus* mandible stress difference | *Varanus* mandible stress difference percentage |
|---|---|---|---|---|---|---|---|---|
| 1 | 0.006 | 0.008 | 0.297 | 2.471 | −0.002 | 28.57% | −2.174 | 157.08% |
| 2 | 0.425 | 0.927 | 2.256 | 4.714 | −0.502 | 74.26% | −2.458 | 70.53% |
| 3 | 2.231 | 2.603 | 1.829 | 0.958 | −0.372 | 15.39% | 0.871 | 62.50% |
| 4 | 0.875 | 1.166 | 0.751 | 1.378 | −0.291 | 28.52% | −0.627 | 58.90% |
| 5 | 0.699 | 0.056 | 0.038 | 3.756 | 0.643 | 170.33% | −3.718 | 195.99% |
| 6 | 0.091 | 0.605 | 0.031 | 8.552 | −0.514 | 147.70% | −8.521 | 198.56% |
| 7 | 0.658 | 0.924 | 1.433 | 1.939 | −0.266 | 33.63% | −0.506 | 30.01% |
| 8 | 0.523 | 1.002 | 2.421 | 2.612 | 0.479 | 62.82% | −0.191 | 7.59% |
| 9 | 0.025 | 1.128 | 2.885 | 5.116 | −0.903 | 133.48% | −2.231 | 55.77% |
| 10 | 0.012 | 0.002 | 0.414 | 3.604 | 0.01 | 142.86% | −3.19 | 158.79% |

The *Varanus* mandible models were two of the most inconsistent in terms of von Mises stress magnitude, deformation, and particularly maximum principal strain. This is likely a result of the relatively extensive reconstructive work applied to both models due to the missing right ramus, comparable to the *Crocodylus* mandible. Stress distributions were noted for their consistency (Fig. 16), with lower von Mises stress occurring in the surface scanned mandible due to dimensionally thicker and denser rami as a result of surface scan reconstructions. Mean unweighted von Mises stress differed by 112.55% and mesh-weighted von Mises stress differed by 63.78%. Max strain differed by 199.99%.

**Table 8 Mean unscaled displacement values in cm at location 1–10 on *Varanus* FE models.** Mean values were calculated by taking the average of five unscaled displacement values at each location.

| Point | *Varanus* cranium CT displacement | *Varanus* cranium surface scan displacement | *Varanus* mandible CT displacement | *Varanus* mandible surface scan displacement | *Varanus* cranium model displacement difference | *Varanus* cranium model displacement difference percentage | *Varanus* mandible displacement difference | *Varanus* mandible displacement difference percentage |
|---|---|---|---|---|---|---|---|---|
| 1 | 0.0115 | 0.0016 | 0.0306 | 0.0373 | 0.0099 | 152.22% | −0.0067 | 19.73% |
| 2 | 0.0092 | 0.0099 | 0.0327 | 0.0838 | −0.0008 | 7.81% | −0.0511 | 87.73% |
| 3 | 0.0131 | 0.0131 | 0.0329 | 0.1190 | −0.0016 | 13.01% | −0.0861 | 113.36% |
| 4 | 0.0156 | 0.0156 | 0.0279 | 0.1410 | −0.0012 | 8% | −0.1131 | 133.93% |
| 5 | 0.0144 | 0.0116 | 0.0102 | 0.1790 | −0.0003 | 2.62% | −0.1688 | 178.44% |
| 6 | 0.0133 | 0.0118 | 0.0080 | 0.1920 | 0.0005 | 4.15% | −0.1840 | 183.96% |
| 7 | 0.0123 | 0.0127 | 0.0205 | 0.2140 | −0.0003 | 2.39% | −0.1935 | 165.03% |
| 8 | 0.0124 | 0.0128 | 0.0216 | 0.2620 | −0.0015 | 12.45% | −0.2402 | 169.54% |
| 9 | 0.0098 | 0.0105 | 0.0197 | 0.1860 | −0.0007 | 6.39% | −0.1663 | 161.69% |
| 10 | 0.0049 | 0.0034 | 0.0133 | 0.1080 | 0.00149 | 36.21% | −0.0947 | 156.14% |

**Table 9 Mean von Mises stress values (GPa) at locations 1–10 on *Chelonia* FE models.**

| Point | *Chelonia* cranium CT scan stress | *Chelonia* cranium surface scan stress | *Chelonia* mandible CT scan stress | *Chelonia* mandible surface scan stress | *Chelonia* cranium model stress difference | *Chelonia* cranium model stress difference percentage | *Chelonia* mandible stress difference | *Chelonia* mandible stress difference percentage |
|---|---|---|---|---|---|---|---|---|
| 1 | 0.314 | 0.138 | 0.983 | 1.195 | 0.176 | 77.88% | −0.212 | 19.47% |
| 2 | 1.693 | 0.649 | 7.959 | 7.795 | 1.044 | 89.15% | 0.164 | 2.08% |
| 3 | 0.818 | 1.029 | 3.814 | 1.156 | −0.211 | 22.85% | 2.658 | 106.96% |
| 4 | 0.651 | 0.935 | 1.539 | 0.869 | −0.284 | 35.81% | 0.67 | 55.65% |
| 5 | 0.659 | 0.669 | 0.985 | 1.379 | −0.01 | 1.51% | −0.394 | 33.33% |
| 6 | 0.871 | 0.731 | 2.403 | 1.464 | 0.14 | 17.48% | 0.939 | 48.56% |
| 7 | 0.564 | 1.124 | 3.191 | 3.572 | 0.56 | 66.35% | −0.381 | 11.27% |
| 8 | 0.456 | 0.802 | 2.648 | 2.263 | −0.346 | 55.01% | 0.385 | 16.67% |
| 9 | 0.419 | 0.244 | 2.962 | 3.921 | 0.175 | 52.79% | −0.959 | 27.87% |
| 10 | 0.243 | 0.776 | 1.224 | 1.848 | −0.533 | 104.61% | −0.624 | 104.61% |

Our specific point mean von Mises stress values overall differed by 99.57% and the mean unscaled displacement values overall differed by 136.95% (Fig. 17; Table 8). Like the other models, stress distributions were noted for their consistency despite the models having the highest unweighted von Mises stress and maximum strain differences.

### *Chelonia* results

The *Chelonia* crania were relatively consistent in their geometric reconstructions, though the bony interior of the skull was difficult to accurately model in the surface scanned version (Fig. 18). Mean unweighted von Mises stress differed by 106.73% and mesh-weighted von Mises stress differed by 11.59%. Maximum principal strain differed by 187.25%. Our

**Table 10 Mean unscaled displacement values in cm at points 1-10 on *Chelonia* FE models.** Mean values were calculated by taking five unscaled displacement values at each location.

| Point | *Chelonia* cranium CT displacement | *Chelonia* cranium surface scan displacement | *Chelonia* mandible CT displacement | *Chelonia* mandible surface scan displacement | *Chelonia* cranium model displacement difference | *Chelonia* cranium model displacement difference percentage | *Chelonia* mandible displacement difference | *Chelonia* mandible displacement difference percentage |
|---|---|---|---|---|---|---|---|---|
| 1 | 0.0272 | 0.0747 | 0.0578 | 0.0497 | −0.0475 | 93.23% | 0.0081 | 15.07% |
| 2 | 0.0244 | 0.0651 | 0.1660 | 0.0886 | −0.0407 | 90.95% | 0.0774 | 60.80% |
| 3 | 0.0147 | 0.0390 | 0.1840 | 0.1660 | −0.0243 | 90.50% | 0.0180 | 10.29% |
| 4 | 0.0128 | 0.0264 | 0.1450 | 0.1020 | −0.0136 | 69.39% | 0.0430 | 34.82% |
| 5 | 0.0214 | 0.0537 | 0.1170 | 0.0579 | −0.0323 | 86.02% | 0.0591 | 67.58% |
| 6 | 0.0237 | 0.0622 | 0.1350 | 0.0622 | −0.0385 | 89.64% | 0.7728 | 73.83% |
| 7 | 0.0268 | 0.0731 | 0.1540 | 0.0703 | −0.0463 | 92.69% | 0.0837 | 74.63% |
| 8 | 0.0123 | 0.0247 | 0.1740 | 0.1320 | −0.0124 | 67.03% | 0.0420 | 27.45% |
| 9 | 0.0184 | 0.0449 | 0.1690 | 0.1270 | −0.0265 | 83.73% | 0.0420 | 28.38% |
| 10 | 0.0251 | 0.0648 | 0.0635 | 0.0547 | −0.0397 | 88.32% | 0.0088 | 14.89% |

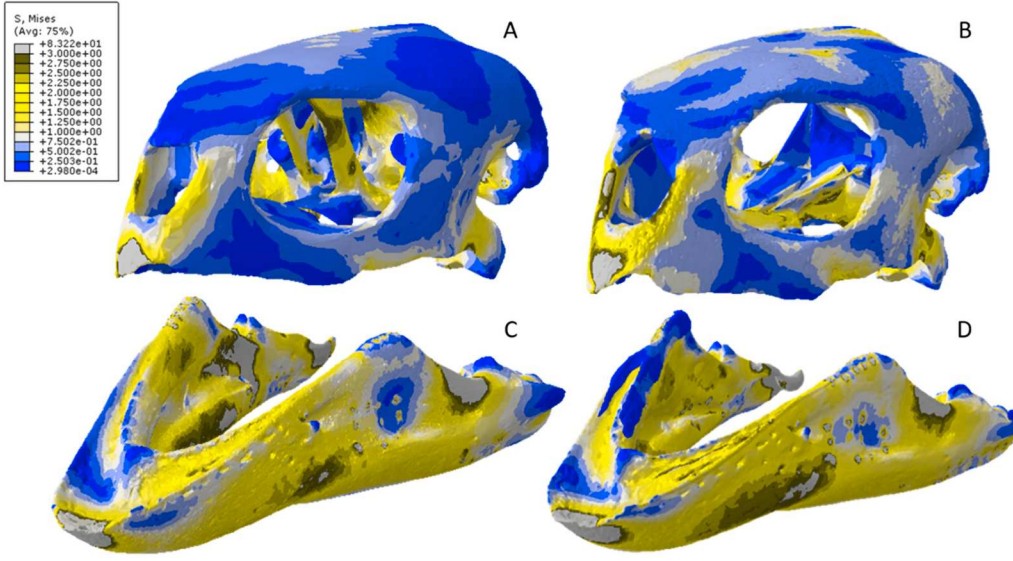

**Figure 18 Von Mises stress results for the *Chelonia* models.** (A) CT scanned cranium, (B) surface scanned cranium, (C) CT scanned mandible, and (D) surface scanned mandible. All FE model images were scaled to the same maximum stress values for consistency.

specific point mean von Mises stress values overall differed by 52.34% (Table 9) and the mean unscaled displacement values overall differed by 85.15% (Fig. 19; Table 10).

Mean unweighted von Mises stress differed by 18.31% and mesh-weighted von Mises stress differed by 6.24%. Max strain differed by 14.79%. Our specific point mean von Mises stress values overall differed by 38.42% (Table 9) and the mean unscaled displacement values

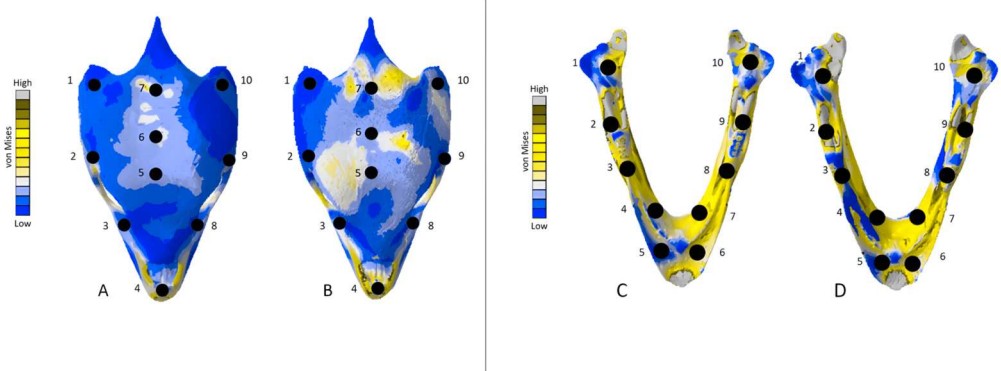

**Figure 19 Dorsal view of the *Chelonia* CT scanned cranium (A) and mandible (B) and surface scanned cranium (C) and mandible (D).** The mean von Mises stress of five elements was calculated at each point and recorded in Table 11. The mean unscaled displacement of five elements to represent deformation was recorded at each point, averaged and documented in Table 12. FE model images were scaled to the same maximum stress value for consistency.

overall differed by 40.77% (Fig. 19; Table 10). The pattern and intensity of deformation was visually identical in both models (Fig. 14).

## DISCUSSION

This study demonstrated that 3D FE results can vary significantly between CT scanned models and surface scanned models, though the distributions of stress/strain occurring in both types of models tends to be similar. We can infer from these results that through use of surface scans, the mechanical attributes (overall stress and strain distribution, deformation patterns) of organisms can be confidently studied. However, the magnitude of stress and strain experienced is more difficult to assess. Calculating the mesh-weighted arithmetic mean (MWAM) to correct for element size can mitigate the differences between von Mises stresses in studies using both types of 3D models, as evidenced in our study.

### Significance of reconstructions

As demonstrated by our *Crocodylus* and *Varanus* mandibles, 3D models which have undergone extensive reconstruction tend to differ most significantly in von Mises stress and strain. We attribute this to the extensive reconstructions which occurred in both models to fix the missing bones (Fig. 1). Difficulty in producing an identical model twice, as well as the process of creating interior-filled surface-scan models, resulted in high variability between models in terms of von Mises stress and topography. This is due to a greater likelihood of models created from surface scan-derived data and those based on CT scan data differing due to scanning procedures and reconstruction. The *Crocodylus* mandible was missing a portion of its left ramus, and the *Varanus* mandible was missing its right ramus in its entirety, which necessitated the use of model editing software Blender 2.82 and Geomagic Studio 12 to duplicate the existing ramus, mirror it, and reattach it to the opposite side of the jaws to complete the mandible.

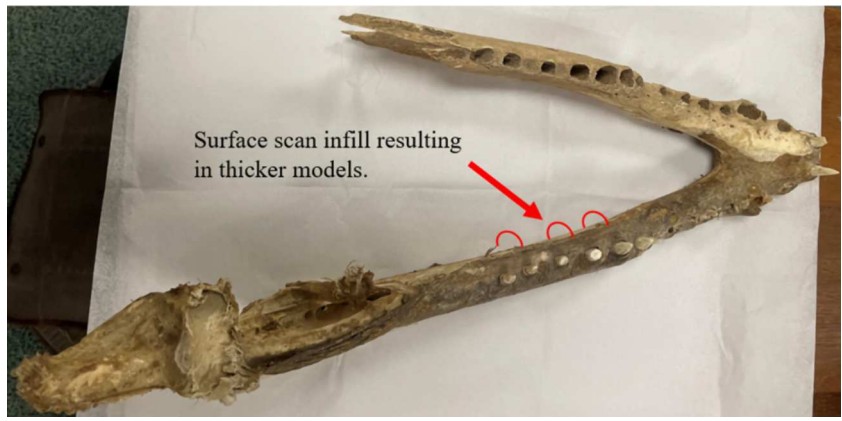

**Figure 20   Dorsal view of the Nile crocodile mandible pinpointing areas of infilling during the surface scan reconstructions.** Fibrous material remaining on the mandible is the main cause of the surface scanned models being denser than the CT scanned version, as the infilling process connected fibrous tissues together and created a larger model than the CT version. This infilling process may also apply to extinct taxa where matrix may still be attached to the fossil rather than soft tissue.

The *Crocodylus* mandible models experienced the greatest discrepancies in von Mises stresses, which we attribute to the extensive editing procedures including duplication and mirroring that can be difficult to precisely reproduce in separate models. The presence of fibrous tissues in the *Crocodylus* crania and mandible also contributed to inconsistencies in surface model generation, leading to further geometric differences between the two models (Figs. 15 and 20). These reconstructive procedures tend to be common in biomechanical studies of fossil specimens (*Nieto et al., 2021*), as most specimens are missing details comparable to the missing bones in this study.

The left *Varanus* mandibular ramus was similarly duplicated and attached at the anterior symphysis; however, the smaller size and geometric simplicity made the process of producing more identical models easier than the *Crocodylus* mandible. We attribute the lower von Mises stress occurring in the surface-scanned mandible to the dimensionally thicker and denser rami as a result of surface scan reconstructions. The right maxillary and jugal bones of the *Varanus* cranium were similarly duplicated and applied to complete the entire cranial model. The relatively low von Mises stress discrepancies between these models may be due to the overall minimal reconstruction necessary in fixing the skull.

Due to its relatively small size, simple geometry, and completeness, the *Chelonia* mandible required the least extensive reconstruction efforts for both CT and surfaced scanned models. The *Chelonia* mandibles also exhibited the smallest discrepancies between model types in terms of von Mises stress and principal strain. We attribute these similarities in FE output to the factors outlined above, which are sharply contrasted by the *Crocodylus* and *Varanus* mandibles. Stresses in the surface-scanned *Chelonia* cranium were more noticeable at the crown of the skull, due to the bony interior being better preserved in the CT scanned model and thus lessening the stresses occurring in bone-laden areas of the model.

The *Chelonia* mandible models were notable as they differed the least out of all models in terms of stress, strain, and deformity, due to the geometrically simple shape and small size requiring minimal reconstruction in both models (Figs. 18 and 19). Generally, models which required the least amount of reconstruction yielded stress, strain, and deformation results that did not deviate markedly between CT scanned and surface scanned versions. However, model simplicity is not a strict requirement for stress and strain congruence, as evident in the *Crocodylus* cranium, which were the largest models by surface area and volume and the second largest in terms of element number but still relatively consistent in FE output.

While this study addressed the question of reconstruction significance in broken and incomplete specimens, which is often the case in biomechanics studies of fossils, there remains the question of whether more complete and simple models could have been utilized first. These models could be used as proof of concept, and function in a similar study of CT scans and surface scans. We did not use hypothetical perfect models, as this study was intended for biological specimens and such incongruences are generally unavoidable in biological studies, especially using fossils.

## CONCLUSIONS

When their utility in 3D FEA studies is compared to CT scans, white light surface scans are effective in capturing deformation and stress and strain distributions. These aspects relate to overall mechanical behaviour and make surface scan models fine candidates for use in studies concerning questions of relatedness in biomechanical patterns. However, surface scans may have questionable results when analysing absolute magnitudes of stress and strain in 3D models. As demonstrated in this study, geometrically simple objects requiring minimal editing, such as the *Chelonia* mandible, will not differ much from their CT versions, especially when the MWAM is calculated. Complex objects requiring little editing, such as the *Crocodylus* skull, also produce comparable results between surface scan and 3D. Objects which require extensive reconstructions, such as the *Crocodylus* and *Varanus* mandible, will result in incongruent absolute magnitudes, though the MWAM calculation still aids in bridging the gap between results.

Studies utilizing both types of scans should attempt to avoid using specimens requiring extensive reconstructive work if possible, *e.g.*, those missing skeletal elements. When this is not possible, extra care must be taken to ensure that reconstructions are as accurate as possible. MWAM calculations are recommended for all comparative FEA studies attempting to compare stress magnitudes in different model types. When this correction was applied, only the *Crocodylus* cranium demonstrated an increase in von Mises stress discrepancy. As the geometries of models created via different scanning methods will vary, these calculations are integral to studies assessing biomechanical attributes of different scan types.

### Future work

This study used surface scanned models that were solidified post-surface reconstruction using the segmentation tools in Avizo Lite 9.7, as surface scanned models are initially hollow

upon creation in Artec Studio 14 Professional. A question remains concerning the validity of hollow surface scanned models and how much they deviate from solidified models in terms of von Mises stress. Studies only requiring the exterior of 3D structures, such as geometric morphometrics, benefit from the time saved in retaining the hollow interior of the models. However, the results of hollow surface scanned models in FE studies and the degree to which their FE output would differ from solid models is not well understood. von Mises stress distributions in hollow models may be similarly worth considering.

This study quantified differences in FE output when comparing different 3D models under identical parameters. One of the difficulties of this study was maintaining identical parameters in both sets of models due to incongruences in model geometry, reconstructions, and muscle nodes. Future work may attempt to compare more geometrically simple models as to limit these inconsistencies between model output. Geometry of our models was kept as consistent as possible; however, variance between models including element count and volume is generally impossible to avoid. Future work may also attempt to refine our results by applying more biologically complex and accurate modeling, particularly making use of more recent muscle data (*Gignac & Erickson, 2016*; *Sellers et al., 2017*; *Wilken et al., 2019*).

We chose not to test simple models, as such models are generally unrealistic in biological studies, and such work may veer more into mechanics rather than biology. The FE-models presented here reflect the nature of the complex geometry of the skull, which does influence FE-model outputs from CT versus surface scanned models. We may infer that models with few inconsistencies will output the most similar FE results.

We analysed von Mises stress and deformation occurring at similar points on CT scanned models and their surface scanned counterparts to quantify deformation between model types. We considered using random points, but there is a risk of those points only landing on very low or high stress areas, and the test may be less informative if not comparing a range of differently-stressed points. Future work may improve upon this method by evaluating FE results using random points and across a larger specimen sample size.

As we noted in our study, the mesh-weighted arithmetic mean (MWAM) is a powerful method of mitigating von Mises stress differences between CT scanned models and surface scanned models. Future work may attempt to further assess the effectiveness of the MWAM in biomechanical studies involving 3D models, particularly those using different types of scans.

## ACKNOWLEDGEMENTS

We thank Elizabeth Martin-Silverstone for CT scanning the reptile specimens. We also thank Claudia Hildebrandt for loaning us the crocodile and monitor lizard skulls from the Wills Memorial Building and Sue Holwell for loaning us the turtle skull from the University of Bristol's Life Sciences Building.

### Funding

NERC grant NE/P013090/1 was awarded to EJR. The funders had no role in study design, data collection and analysis, decision to publish, or preparation of the manuscript.

### Grant Disclosures

The following grant information was disclosed by the authors:
NERC: NE/P013090/1.

### Competing Interests

The authors declare there are no competing interests.

### Author Contributions

- Andre J. Rowe conceived and designed the experiments, performed the experiments, analyzed the data, prepared figures and/or tables, authored or reviewed drafts of the article, and approved the final draft.
- Emily J. Rayfield conceived and designed the experiments, analyzed the data, authored or reviewed drafts of the article, and approved the final draft.

### Data Availability

Muscle force calculations for the crocodile, monitor lizard, and turtle are available in the Supplementary File.

The 3D scans are available at MorphoSource:
- http://dx.doi.org/10.17602/M2/M432013
- http://dx.doi.org/10.17602/M2/M432009
- http://dx.doi.org/10.17602/M2/M432005
- http://dx.doi.org/10.17602/M2/M432001
- http://dx.doi.org/10.17602/M2/M431997
- http://dx.doi.org/10.17602/M2/M431987
- http://dx.doi.org/10.17602/M2/M432175
- http://dx.doi.org/10.17602/M2/M432179
- http://dx.doi.org/10.17602/M2/M432183
- http://dx.doi.org/10.17602/M2/M432187
- http://dx.doi.org/10.17602/M2/M452506
- http://dx.doi.org/10.17602/M2/M452562
- http://dx.doi.org/10.17602/M2/M452652
- http://dx.doi.org/10.17602/M2/M452655
- http://dx.doi.org/10.17602/M2/M452688

### Supplemental Information

Supplemental information for this article can be found online at http://dx.doi.org/10.7717/peerj.13760#supplemental-information.

# PeerJ

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
