# Peer review of "The efficacy of computed tomography scanning versus surface scanning in 3D finite element analysis"

_PeerJ, doi:10.7717/peerj.13760_

## Round 0.1 · original submission · Major Revisions

· Academic Editor

Major Revisions

The reviewers clearly think this can be an important contribution and I agree. Please read through the reviewer comments carefully and make changes as appropriate. Make certain to be detailed in your resubmission letter, especially where you think reviewer concerns were unwarranted. I'm excited to see your revised manuscript.

Reviewer 1 ·

Basic reporting

1. Lines 63 to 65: The sentence contained here states one fact (“typically pertain to…”) then adds another fact (“studies have focused…); as written they sound disjointed, possibly because of the tenses. It appears that the authors’ intent was to imply that whereas argument A is typical, argument B also exists but it does not read that way and instead reads argument A is typical, argument B existed.
2. Line 101: “scanning or photogrammetry” is introduced here as equivalent, but this is mid-paragraph. Should this not be made equivalent earlier so that this entire section can relate to both methods?
3. Line 104 and 105: The authors switch between use and utilize in consecutive sentences (and in a single sentence later – line 233). I would argue for consistent application of one word or the other throughout the manuscript.
4. Line 129: The point has been made that CT scanning is more expensive at least once prior; is it necessary again here?
5. Line165 – 167: The definitions contained in these lines are easy to read and easy to follow. They are really nicely conveyed and should be contained in future FEA papers to give readers new to the topic a common ground with which to approach stress and strain in scientific papers. Very well done.
6. Line 320: See my note below (in general comments, point 3) about consolidating tables if possible as referring to “Tables 1, 2, 3, 4” may look a little daunting to the reader.
7. Line 392: Instead of saying an animal’s “CT scanned morphology” could you say the models deviate from typical morphology? The sentence as written gives the impression that CT scanned morphology is different from what we might expect of a living animal. If it is, then no computer model derived from CT data would be relevant to a living animal, correct?

Experimental design

1. Line 179: This section begins by delving directly into the first specimen. In a later line (L185-186) a second specimen is mentioned that has not yet been introduced. Perhaps it is worth adding a sentence at the beginning of this section to remind readers of the three specimens being examined. This could also provide a place to identify specimen numbers if they exist.
2. Line 189-191: I assume choosing incomplete specimens helps to elucidate difficulties with fossil modeling and was purposely (as alluded to in the next sentence) done to mimic this. Assuming this was done for this purpose, these are good choices. Some readers may ask why the authors would not look for complete specimens, and it may be pertinent to provide an example or two here or elsewhere that supports your choices; Nieto et al. 2021 (Araripesuchus modeling) comes to mind as a good example.
3. Line 193-194: Are there any difficulties in segmentation or surface scan modeling that may be associated with the smaller taxa because they were scanned in articulation whereas the crocodile was disarticulated? This may be answered later, but I will leave this comment here as other readers may have the same question in this portion of the paper.
4. Line 197: What is meant by scaled down here? File size? Measurements of the in-silico model?
5. Line 202: Is it worth briefly explaining (an added clause to this sentence could be sufficient) what the “‘real-time fusion’ option” tool does?
6. Line 203: Here all elements were separately scanned (due to file size, it is noted). This may bring up the question posed in Point 3 above. Why were they scanned in one configuration in one portion of the study but a different way in the other portion? We could all agree the model would (likely) be hopelessly intertwined if they were articulated while surface scanning, but unfamiliar readers may need this point briefly described.
7. Line 210-212: The background reasoning described here is the sort of statement/explanation that would be helpful with Point 5 above. This is very well worded and described.
8. Line 214-215: Can a definition be provided for the “acceptable quality” for either Blender or Geomagic as a model destination?
9. Line 231: There is no mention of the missing (portion of?) pterygoid and/or generally fragmented appearance of the cranium of Varanus. How was this reconstructed in the model? Is it the same process?
10. Line 254: Some readers may ask why anterior (weaker) bite points rather than posterior (stronger) bite points were chosen. Is there reasoning available for this and would it make sense to add this to this paragraph? A related question to this is the missing depressor muscle. Though not necessarily active in biting, might its antagonistic contraction on the opposite side of the jaw joint mitigate some of the force introduced into the system during a bite?
11. Line 263: The authors may want to look at Wilken et al. 2019 to use an updated Varanus muscle model which changes insertion points for at least one muscle in varanid lizards. Additionally, Gignac and Erickson 2014 and Sellers et al. 2017 may provide updated crocodilian muscle maps worth investigating.
12. Line 292: Is there a specific package in R used to calculate MWAM? This should be shared or stated if one exists, otherwise there should be a brief description of how this is done, if feasible, to help others reproduce these results in the same fashion.

Validity of the findings

1. The reporting of the findings is really well done. I initially had a question about what solid filled surface scanned models means, and I left it in the comments below (Point 2 in additional comments – I still have questions that are unanswered there), but I do see in the “Future work” section that this is partially addressed, which I appreciate. I concur that now a comparison should be done between CT, hollow, and filled surface scanned models to validate the findings here and usefulness, or not, of filled versus hollow models.
2. Additionally, I appreciate that the reconstruction that occurred is addressed by the authors, not only from a fossil reconstruction standpoint (as in, this is a reality for fossil modelers and so it needs to be investigated), but also a “broken specimen” standpoint. I do wonder though, if more “perfect” models should have been tested first using these methods, as a proof of concept. Could a short statement be appended to the introduction or discussion/conclusion explaining why the authors chose a “not quite worst” case scenario as the first test case?
a. In the future work section this is briefly mentioned (Line 454-455), but not explained. This still begs the question, why not show the most simple comparison first?

Additional comments

1. In the methods section it is mentioned that each model has the number of triangles reduced. It is then noted in the results sections that there was a difference in the number of elements and the percentage difference was reported. After articulation of the surface scanned and CT models, would it not be possible to set the number of triangles to an equal amount and still preserve the trabecular architecture gleaned from the CT scanning?
2. In the methods and again in the results you mention creating interior-filled surface models. Are these models simply filled in, amounting to a large increase in tetrahedral elements, or are they meant to mimic the trabeculae observed in the CT scanned models? I thought I knew what you were describing, but I want to make sure, as it does seem a little ambiguous and I think other readers may also be a little confused.
3. There are many tables and figures. The information in these tables and figures is very important, but I wonder, for example, if the authors might be able to combine tables 1 through 4 into a single table. Likewise, could some of the figures also be combined to better use space? As an example, could figures 6 through 8 become one figure with all of the muscle maps placed together with a single color-based legend? I have similar thoughts/questions about combining the FEA result figures (e.g. merging 14, 17, and 20 OR 14-16, etc. could be viable ways of making multiple figures a single figure).
4. Related to Figures 6 – 8: Why are they in a different order from the skull and FEA presentation (Crocodylus, Varanus, Chelonia – here it is Varanus, Crocodylus, Chelonia)? Also, there are multiple different orientations shown here. I understand this was done to best show the attachments, but I think it would be better to have some consistency across the figures. As they stand now, they all present different views and even the oblique mandibles are all at different oblique angles. This requests more effort from the reader to read each figure. Regarding the previous point, merging these into a single figure would require that they all be similarly aligned, otherwise they would be highly disjointed and confusing.
5. Figure 17 has a ghost image in my copy of the manuscript. This may need to be double checked.
6. Throughout, but starting at Line 315: MWAM was already explained, so I don’t believe that it needs to be written out again.
7. Typos or awkward wording:
a. Line 116: Should read “is a sufficient method…”.
b. Line 217: Typo at the end of the sentence.
c. Line 284: Typo at the beginning of the sentence.
d. Line 293: Typo in “athematic”; my guess is that it should read “arithmetic”.
e. Line 395 & 402: Use of rami to refer to a singular element (instead of ramus).
f. Line 418: Awkward wording toward the end of the line.
g. Line 442: “This studied” should be fixed.
h. Figure 16 is never referred to in the text.

·

Basic reporting

There are numerous punctuation, grammar, and other issues with the text of this manuscript in addition to several typos present throughout the text. In addition, there are several sentences that, while technically acceptable, are far too long. In a scientific manuscript, a single sentence that is 4 or more lines long often ends up being confusing because by the time the reader has reached the end of the sentence, the beginning of the sentence has been forgotten and point being made has been lost. I have pointed out a very small number of issues in the general comments of this review, but I would strongly recommend at least one more proofreading of this manuscript by each of the authors and another outside party to help identify and correct each of these problems. No writing is error-free, but such issues should be minimized to avoid letting the text become confusing, which is what happens to the current manuscript in a handful of places.

The first paragraph of the “Surface scanning” subsection, despite being part of the Introduction, contains no citations. Such an omission is a problem since that paragraph, as part of the Introduction, includes several sweeping statements and assertions about methodology and trends in the literature without reference to examples of such literature.

Conversely, lines 183-188 and line 199 of the Materials & Methods section include explanations of why specimens were sampled and data were formatted in specific ways, along with citations for those passages. Such explanations would be better suited in the Introduction, where they provide background to justify the study methodology, or the Discussion, where the explanations they provide for the authors’ decisions can address potential concerns a reader may have with the chosen procedures. In general, methods should not include citations because they are a chronicle of events, not a history complete with context and explanations.

Figure 5; the yellow text is practically illegible. The main culprit for that illegibility is the extreme overlap of nearly all of the labels with one another. Even with the problem of text overlap solved, the combination of font size and color, especially with the model background being green and the figure background being white, would probably still make those labels very difficult to read. Since the point of this figure seems to be illustration of applied joint and bite point constraints, and because the muscle attachment mapping is illustrated much better in Figure 7, the labels for muscle attachment locations can probably be removed from this figure, and the labels for the constraints can be more clearly labeled.

Table 3; there is no unit given. I assume these volumes are in cubic millimeters, but that is just a guess based on personal experience that very few readers will have.

Table 4; no unit given.

In general, this manuscript relies pretty heavily on the background knowledge of the reader to be able to follow along and make sense of several passages, particularly in the Material & Methods section. Many statements are made throughout the text without a level of explanation sufficient for a wider audience and/or citation so readers can seek out additional information themselves. For example: why is the preservation of fibrous material in the C. niloticus skull a potential issue when surface scanning (Lines 181-183; 399-401)? Another example is that the surface scan-based models are repeatedly described as “hollow,” whereas other passages refer to them as being more dense than their CT data-derived counterparts. One of the explanations for the increased density is the presence of soft tissues which, presumably (though not explicitly stated), was virtually removed from the CT based-model during segmentation but not from the surface scan-based model. In that case, the models vary in thickness or volume, not density. Perhaps the soft tissues include fibrous material within the cranial sutures, which would seem to make the surface scan-based model more solid than the CT-based model that would represent more disjointed bony elements. I can think of a few other possibilities for why the preserved soft tissues may impact the surface scan-based models but not the CT-based ones, yet none of them would alter the density since, as you stated, the surface scan-based models are hollow until you fill them. Several similar examples can be found throughout the manuscript text of other ways that the authors seem overly reliant on the ability of the reader to fill in information gaps, make logically leaps, and otherwise possess sufficient specific background information and experience to understand some key points on their own. I encourage the authors to examine the text for such examples and, when they are found, make sure that methodological procedures, interpretations of the results, and so on are clearly, explicitly, and fully described and explained.

This study and report is sufficiently self-contained, and the reported results are relevant to the stated hypotheses. Appropriate raw data are available with the article, and additional data (such as the raw CT scan data) are presumably available upon request since there is no mention of an online repository where those data are stored and available for download.

Experimental design

This manuscript reports the results of original primary research within the Aims and Scope of this journal.

The research question is well-defined, relevant, and meaningful. Although some aspects of the premise of this study seem intuitive, ground-truthing such items so they meet the threshold of scientific rigor upon which future studies may be built is extremely important. The research presented in this manuscript is particularly important given that biomechanical FEA research is becoming more widespread. That growth in this type of research makes studies such as this one necessary to identify potential pitfalls and establish appropriate protocols for future research.

The investigation described in this manuscript appears to have been performed to the highest technical standard currently widely available. There are other ways of addressing the study questions presented, such as using a simple model to represent a generalized skull or other morphology, rather than models derived from scan data of actual biological specimens, but the decisions and methods of the authors are valid for this study.

The methods are not currently described with sufficient detail and information for another researcher to reliably replicate. See my comments in the “Basic Review” section of this review and my specific point in the “General Comments” section below. I believe that another researcher within the authors’ research group could replicate this study with minimal assistance, but I do not believe that I, as an outside party, could replicate this study without making assumptions and guesses about what to do and not do at certain points. Although I could probably guess the correct option in most or all of those instances, another less experienced researcher, such as a future student, would almost certainly have difficulty without considerable guidance and assistance. Given that the importance of this study is in its value for helping establish protocols for obtaining consistent results across studies, not clearly, explicitly, and fully describing and explaining all aspects of the methodology reported in this manuscript does not uphold the promise of this study.

Validity of the findings

The findings of this study appear valid. Some minor aspects of the authors’ conclusions are not entirely intuitive or clearly explained, but I assume that addressing many of the overall issues and specific points listed elsewhere in this review will go a long way to clearing up any confusion about those conclusions. The authors also provide some suggestions for directions for future research.

The underlying data, in the form of CT and surface scan datasets have not been provided, either as supplemental data or in the form of reference to an online repository. Despite the problems such lack of open availability pose for other researchers, it is very far from uncommon for researchers in the authors’ field to store such digital datasets in personal collections and make those data available upon request, which is what I assume is the case here. Those scan data, as described by the authors, appear to have been appropriately obtained and analyzed.

Additional comments

Lines 93-102 without citations

Lines 127-129 is an important statement yet without citation

Line 152; why was such resolution unlikely achieved

Line 196; what does voxel size 1 mean? Is that the default 1mm voxel setting (which wouldn’t match your data, based on your description of the scan resolution), or does that dimension match the native resolution of the scan data? I’m far more proficient with Avizo than most readers will be, but I don’t know what voxel size 1 is. I am only familiar with datasets that contain voxel dimensions within the data, such as DICOM, or datasets in a format such as a TIF stack that require manual input of voxel dimensions when the data are first opened in Avizo.

Line 197; what does “scaled down” mean? Where the data downsampled, and if so, to what voxel size? Were the surface models generated from the segmented scan data simplified, and if so, how? By a set amount (i.e. to a uniform number of triangles), by a percentage of triangles, or according to some other metric?

Line 207; what constituted satisfaction?

Lines 224-228 are not clear as to the information they are intended to convey.

Line 229; more than just the articular side of the left craniomandibular joint is missing, which seems to be what is meant by “hinge,” and it is missing entirely rather than broken, as indicated. The phrasing of the following sentence is much clearer. If you mean that the posterior end of the left mandibular ramus is not preserved, simply state that.

Lines 240-241; by what criteria and to what extent were the element and triangle counts reduced, beyond satisfying the need to reduce analysis run times? McCurry et al. (2005) found that a surface resolution of 300,000 faces is required to obtain stable FEA results in a comparative context. Thus, reducing triangle counts can be very helpful for shortening analysis run times, but too much reduction may cause results to vary from one analysis to another, even when all model parameters are identical.

Lines 241-243; did you create a uniform tessellation for your surface models prior to meshing? In addition to creating more consistent and repeatable meshes, performing this step with something like the “remesh” tool in Geomagic Studio helps avoid artifacts in FEA results due to irregularly sized (i.e. very small next to very large) triangles, and thus tetrahedral, being clustered next to each other. This step may also be unnecessary since you calculated MWAM, in which case some preview that irregular element sizes within the model meshes were addressed later in the study procedure could be added.

Lines 248-250; it seems that you assigned crocodylian bone properties to both the Crocodylus and Chelonia specimen models, though that’s not entirely clear. This whole sentence would be clearer if you simply and explicitly stated which material properties were assigned to each of the models. For example, I assume the lizard bone properties were assigned to the lizard model, but since that isn’t actually stated, I could be wrong. Currently, it reads as, “lizard bone properties were assigned,” without stating to which model those properties were assigned.

Lines 283-291; how were the sampled points selected? Was some biologically informative regime used, were the locations based on the stress heat map for each model or taxon, were they randomly selected, or was some other basis used? As long as they are consistent between the two models of each taxon the mechanism for selecting those points probably doesn’t matter, but since this is a scientific study, every aspect of the methodology should have some rationale for its inclusion and repeatable procedure for its implementation. That would also help this study have broader impacts beyond a ground-truthing function.

Lines 390-392; I assume the differing you mention is between models based on surface scan-derived data and those based on CT scan data, but because that is not explicitly stated, the intent of the term “differing” is unclear.

Line 395; disagreement between singular and plural terms.

Line 402; disagreement between singular and plural terms.

Lines 452-461; this section appears to contain the same assertions restated multiple times. As a result, the point is a little difficult to tease out of the background noise of redundancy.

Figure 2; caption should specify that the single preserved ramus is of the mandible since many bones exhibit a ramus.

Table 2; see my earlier comment on Lines 204-241. Resampling your surface models in Geomagic Studio can help set your pairs of models to a similar number of surface triangles to one another, which in turn should lead to more consistent element counts and convergent results. Your findings may or may not be altered, but since model resolution is a potentially highly impactful variable, even when comparing multiple analyses of a single model, it should be addressed in some way when comparing pairs of models.

This study is important for those conducting research in this area. As such, I would greatly like to see this manuscript accepted for publication. However, there are several issues with the text that I feel need to be addressed first, many of which are related to the ability of this study to be replicated and expanded upon. Overall, this manuscript has all of the hallmarks of being the first (or at least one of the first) articles written by a student. I hope that my general, broad comments assist the lead author in re-evaluating their writing in the same way that I and many other students have had to do. The specific points I listed in the General Comments section are not intended to be exhaustive but are intended to act as examples of what to look for elsewhere in the manuscript as well. If the authors can write this manuscript in a way that lets other readers follow along and understand it as well as the authors can, it will be a very important contribution to the literature.

---

## Round 0.2 · Minor Revisions

· Academic Editor

Minor Revisions

This is good and ready to be accepted. Please just give a quick look at the editorial suggestions of reviewer 2. Several of them are worth considering. Maybe spend 30 minutes on these minor considerations and it is good to go.

Reviewer 1 ·

Basic reporting

The authors have answered all of my questions and comments with adequate responses. I think the additions and deletions as well as the altered text have created a better manuscript that is suitable for publishing. Small details were added throughout that make tools and software decisions easier to comprehend and to use outside of this study (such as the addition of R code that was used). The only question I have relating to this code snippet is whether it should be indented as if a quote or mathematical equation.

Experimental design

The additions to the experimental design provide solid reasoning behind choices made by the authors and this, in turn, bolsters the paper. As currently written, the design is more repeatable than the previous version. It also allows for more understanding and/or rebuttal and informed suggestion from peers. The additions are extremely useful and informative.

Validity of the findings

The new portions of the background and experimental design help to shore up the reasoning previously detailed in the findings. New information that has been added here addressing reviewer concerns also helps to better explain and identify the findings. The findings were previously valid but needed a little more explanation or added information. This has been done in this version of the manuscript and I find the additions to be of value to the overall paper.

Additional comments

I think that this is an interesting and very important study. I look forward to the authors' future works as well. Thank you for allowing me the pleasure of reading these early versions!

·

Basic reporting

The text of this manuscript is generally clear and unambiguous, and professional English is used throughout. However, several typos, grammatical issues, and a small number of less-than-clear statements are still present. Such instances were noted in the annotated PDF file attached to this review, though some issues that appear multiple times were identified only a few times with the hope that the remainder could be identified and corrected by the author.

References, background, and context are all largely sufficiently provided. However, some statements in the Introduction are still unsupported. Such instances were noted in the annotated PDF file attached to this review.

The figures are greatly improved over the previous version, tables are appropriate and clear, and appropriate raw data has been shared. The current structure and organization of the manuscript could be acceptable, but some reorganizing and regrouping of certain passages in the Results, Discussion, and Conclusions section would greatly benefit the flow and clarity of the points and arguments presented. Examples of such passages were noted in the annotated PDF file attached to this review.

This article is self-contained, and results are relevant to the stated hypotheses.

Experimental design

The research described is original, meaningful, and sufficiently described so as to be replicated with relative ease.

Validity of the findings

The novelty of the reported findings was clearly identified, possible methods for future replication and elaboration were suggested and encouraged, underlying data were provided and appear robust, and the conclusions were both clearly stated and limited to supporting results.

Additional comments

This revised manuscript is considerably more mature than the previous iteration. It could potentially be published as-is. However, there are enough remaining issues with typos, repetitive statements, and so on, that I feel a little more polish would result in a manuscript of which the author could be considerably prouder. Minimally, some relatively minor reorganization by moving a few statements between the Results, Discussion, and Conclusions section would greatly improve the flow of the manuscript and clarity of the arguments being made. Doing so would also help future researchers find the information they need in the appropriate sections when reading and referencing this work.

At present, I feel this manuscript does not fully live up to its potential, largely because of the aforementioned disorganization between sections. The text does not necessarily fail to meet PeerJ criteria, but I believe that one more round of revisions by the author would immensely help them get their point across and help the audience make sense of the text as a whole. However, if the authors and/or editor object to revisions of that sort, I can accept such a decision as long as the typos, misspellings, and other textual errors are corrected prior to acceptance. With or without reorganization of the text, I do not strongly feel another round of reviews is necessary.

---

## Round 0.3 · accepted · Accept

· Academic Editor

Accept

Nice job. Look forward to seeing this published!